# LARGE LANGUAGE MODELS COULD BE ROTE LEARNERS

## ABSTRACT

Multiple-choice question (MCQ) benchmarks are widely used for evaluating Large Language Models (LLMs), yet their reliability is undermined by benchmark contamination. In this study, we reframe contamination as an inherent aspect of learning and seek to disentangle genuine capability acquisition from superficial memorization in LLM evaluation. First, by analyzing model performance under different memorization conditions, we uncover a counterintuitive trend: LLMs perform worse on memorized MCQs than on non-memorized ones, indicating the coexistence of two distinct learning phenomena, *i.e.*, rote memorization and genuine capability learning. To disentangle them, we propose **TrinEval**, a novel evaluation framework that reformulates MCQs into an alternative trinity format, reducing memorization while preserving knowledge assessment. Experiments validate TrinEval's effectiveness in reformulation, and its evaluation reveals that common LLMs may memorize by rote 20.5% of knowledge points (in MMLU on average).

## 1 INTRODUCTION

The rapid advancement of Large Language Models (LLMs), driven primarily by large-scale pre-training on massive datasets, has endowed these models with remarkable proficiency across diverse tasks (Ouyang et al., 2022; OpenAI, 2024; Touvron et al., 2023). As LLMs continue to improve, evaluating their genuine capacities has emerged as one of the fundamental challenges, necessitating proper methodologies to ensure fairness and robustness (Ganguli et al., 2023; Liu et al., 2023b).

Among the developed methods, multiple-choice question (MCQ) benchmarks have become a standard approach for evaluation. Typically, LLMs are presented with a question and a fixed set of answer choices, requiring them to select the most appropriate option (see Fig. 1 for illustration). This format enables straightforward performance measurement through accuracy metrics and could cover a wide range of subjects. However, despite their widespread adoption, MCQ-based evaluation raises concerns about reliability

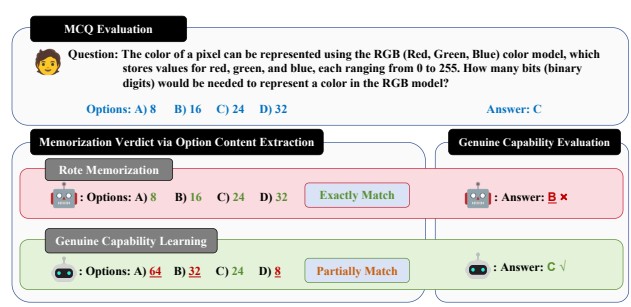

Figure 1: MCQ-based evaluation. We observe that LLMs tend to underperform on memorized MCQs.

due to benchmark contamination (Li & Flanigan, 2024; Kim et al., 2024), *i.e.*, test data unintentionally appears in training corpora and models may exploit memorized content rather than demonstrating genuine understanding, inflating their apparent capabilities. For instance, Zhou et al. (2023) discovers that smaller models with deliberate pre-exposure could outperform their larger counterparts, thereby contradicting widely accepted scaling laws.

To mitigate the issue, Zhou et al. (2023) advocates the removal of benchmark datasets from pre-training corpora. However, this strategy conflicts with the fundamental objective of large-scale pre-

training, which aims to maximize model performance by exposing LLMs to as much data as possible. From a broader perspective, human learning also involves problem-solving through practicing on similar questions, *e.g.*, exam preparation. While rote memorization of specific questions and answers merely lead to short-term success, repeated practicing can also facilitate deeper conceptual understanding. Inspired, rather than viewing benchmark contamination as a flaw to be eradicated, which is a nearly impossible task at scale (Sainz et al., 2023; Bordt et al., 2024), we argue that it is an inherent aspect of learning and should be accounted for in evaluation. Therefore, this study shifts its focus to *evaluating LLMs in the presence of contamination, aiming to distinguish genuine capability gains from superficial memorization effects*. The explicit disentangling of these two learning effects remains largely unexplored in MCQ-based evaluation, yet we believe it marks a crucial step towards developing more rigorous and unbiased evaluation methodologies.

To investigate the effects of superficial memorization in LLM evaluation, we compare model performance under different memorization conditions. Inspired by membership inference attacks (MIA) (Carlini et al., 2022a; 2021), we define superficial memorization as an LLM's ability to verbatim reproduce content, *e.g.*, exactly extracting option contents of MCQs in our case. Using this criterion, we partition the MMLU benchmark (Hendrycks et al., 2020)[1] into memorized and non-memorized subsets and evaluate three open-source LLMs[2] on both. Surprisingly, results reveal a consistent yet counterintuitive trend: LLMs perform worse on memorized MCQs than on those not (see Fig. 1 for illustration and Fig. 2 for results). This challenges the assumption that memorization improves model performance and suggests the coexistence of two distinct learning phenomena in LLMs: *rote memorization*, where models recall content verbatim without true understanding, and *genuine capability learning*, where they internalize underlying knowledge.

The preliminary investigation has several limitations. First, binary classification of MCQs as either memorized or non-memorized oversimplifies the nuances of memorization, potentially overlooking intermediate cases. Second, we rely on accuracy to measure performance, which is inherently unreliable. Third, our analysis could not reveal the numerical dependency between rote memorization and capability learning. To address these challenges, we propose **TrinEval**, a novel evaluation framework designed to provide a more reliable measure of LLM performance by minimizing the influence of rote memorization. TrinEval employs a query-based probing (q-probing) mechanism Allen-Zhu & Li (2023) that reformulates MCQs into an alternative trinity format, *i.e.*, entity-attribute-context. This could prevent direct content recall while preserving knowledge assessment.

Through experiments, we demonstrate that TrinEval's reformulation is knowledge-preserving, *i.e.*, maintaining testing problems' inherent knowledge requirements without introducing extra cues, and could effectively reduce memorization. Combined with a continuous superficial memorization quantification metric, TrinEval reveals the in-robustness of LLMs' capability learning, *e.g.*, with MMLU, tested open-sourced LLMs only mastered 19.6% of knowledge points while 20.5% are memorized by rote in the meanwhile, shedding light on the necessity for further optimization.

## 2 RELATED WORK

### 2.1 LLM EVALUATION ON MCQ BENCHMARKS

The rapid advancement of LLMs has driven their expansion into diverse domains, necessitating robust and fair evaluation methodologies (Zheng et al., 2023b; Hu et al., 2025) and platforms (Contributors, 2023; Chiang et al., 2024). Among these, evaluating on MCQ benchmarks emerges as a widely adopted approach due to the ease of validation and standardized comparison across models (Hendrycks et al., 2020; Wang et al., 2024; Zhong et al., 2023; Huang et al., 2024).

However, MCQ-based evaluations are not without limitations. Biases in LLM responses have been extensively studied (Dai et al., 2024), revealing issues such as social biases (Salewski et al., 2024; Liu et al., 2023a) and order sensitivity (Akter et al., 2023). To mitigate the latter, PriDe (Zheng et al., 2023a) estimates the option positional bias after option permutation. To examine mastery of knowledge, Zhao et al. (2023) applies a hypothesis testing method and checks rephrased-context

---

[1]Selected for its popularity and documented data contamination in widely used LLMs (Sainz et al., 2023).

[2]Llama2-7B (Touvron et al., 2023), Mistral-7B-v0.2 (Jiang et al., 2023) and Vicuna-v1.5-7B (Zheng et al., 2023b).

consistency for a given question. Benchmark contamination is arguably the most severe challenge for MCQ-based evaluations, which may result in misleadingly inflated performance (Zhou et al., 2023; Li & Flanigan, 2024). To address this, prior studies have explored data filtering, frequently-updated test sets (White et al., 2025), and data perturbation (Li et al., 2024).

In this paper, instead of attempting to eliminate contamination, we evaluate LLMs under its presence, aiming to distinguish genuine capability gains from superficial memorization effects. This marks a new perspective of LLM evaluation, revealing the extent to which models truly understand concepts rather than merely memorizing data.

## 2.2 LLM MEMORIZATION

Membership inference attacks (MIA) are commonly used to determine whether a specific sample was present in a model's training data. Initially studied in smaller models, Carlini et al. (2022b) investigates deep learning memorization mechanisms by identifying and removing easily detectable memorized samples. In the context of LLMs, MIA has been employed to assess privacy risks, revealing that both open- and closed-source models can leak sensitive personal data when provided with related prompts (Kim et al., 2024).

Beyond privacy concerns, Carlini et al. (2022a) formally defines LLM memorization as a model's ability to verbatim generate text sequences following a prefix prompt. Using this definition, several studies (Sainz et al., 2023; Bordt et al., 2024; Carlini et al., 2021) have examined mainstream LLMs, confirming widespread test data leakage across popular benchmarks. To quantify memorization strength, researchers (Shi et al., 2023; Zhang et al., 2024; Oren et al., 2023; Carlini et al., 2019) have further explored methods such as analyzing token probability distributions in generated outputs. However, while these studies extensively analyze LLM memorization, few explicitly investigate how memorization influences an LLM's problem-solving ability. In contrast, our work focuses on their interplay, presenting a more rigorous approach to fair and reliable LLM evaluation.

## 3 METHODOLOGY

### 3.1 PRE-INVESTIGATION OF LLM CAPABILITY W.R.T. MEMORIZATION

Benchmark contamination often leads to inflated performance estimate. This phenomenon is commonly attributed to models memorizing specific questions and answers rather than demonstrating genuine problem-solving abilities. However, the extent to which and how memorization influences LLM performance remains unclear. To disentangle genuine capability acquisition from superficial memorization, we conduct a preliminary investigation into how LLMs perform under different memorization conditions. By examining model accuracy on memorized vs. non-memorized subsets, we aim to reveal the role of memorization in LLM evaluation and establish a foundation for more rigorous assessment methodologies.

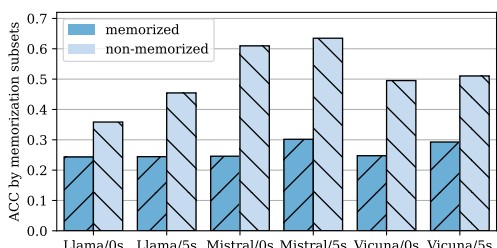

Figure 2: Model performance on memorized and non-memorized subsets of MMLU, where '0s' and '5s' stand for zero- and five-shot prompting, respectively.

In this paper, we mainly discussed the LLM Capability w.r.t. Memorization via MCQ problems as an example. The generalizability of our approach is discussed in the Appendix G. Formally, we define an MCQ as $x = \{x_Q, x_O, x_W\}$, where $x_Q$, $x_O$, and $x_W$ refer to the question, options, and ground-truth answer, respectively. Following the memorization definition from Carlini et al. (2022a), we say an MCQ $x$ is memorized by LLM $G$ if $G$ can extract/generate the content of options $x_O$ exactly given question $x_Q$. In practice, given the commonly used next-token prediction pre-training, we incorporate meta-information (*e.g.*, benchmark name) and 5-shot examples to elicit the memorized following content and recall memory (refer to Appendix E), and use greedy decoding (*i.e.*, temperature fixed to 0) during extraction (Bordt et al., 2024; Sainz et al., 2023) (refer to Appendix A for the complete prompt). Using MMLU (Hendrycks et al., 2020) as the evaluation benchmark, we

divide the test set MCQs into memorized and non-memorized subsets, where the memorized subset consists of 909–982 questions (accounting for 6.5%–7.0% of the total 14,006) depending on the tested LLMs Llama2-7B, Mistral-7B-v0.2, and Vicuna-v1.5-7B. The detailed statistics of questions across subsets are given in Table 1 in Appendix, and we also observe that the majority of memorized questions are those relatively simple, *i.e.*, not in MMLU-PRO (Wang et al., 2024).

We then compute the accuracy (ACC) of tested LLMs by subsets as a proxy of model performance under different memorization conditions. The results of both zero- and five-shot prompting are reported in Fig. 2, from which we observe a consistent yet somehow counterintuitive trend: LLMs exhibit 47.2% lower accuracy on average on memorized MCQs compared to non-memorized ones, regardless of LLMs and prompting techniques. This finding challenges the commonly held assumption that memorization directly improves model performance. In addition, it also implies the coexistence of two distinct learning paradigms within LLMs, which we term rote memorization and genuine capability learning, respectively.

However, our pre-investigation has its limitations. The binary classification of memorization potentially overlooks more nuanced forms of learning. Additionally, using ACC as the performance metric does not truly capture model capacity. We address these two issues in the following subsections, which then ensure a disentangle analysis of rote memorization and capability learning.

## 3.2 QUANTIFYING LLM MEMORIZATION

For quantifying the memorization of LLMs, prior research (Shi et al., 2023; Zhang et al., 2024) suggests that outlier tokens, which exhibit higher generation probabilities, are more likely to be found in memorized samples. Experiments on WIKIMIA (Shi et al., 2023) also finds that this method exhibits high AUC score on seen samples. Building on this idea, we develop a metric that utilizes the bottom $K\%$ of token probabilities within the generated sequence as a measure of memorization (we primarily set the value of $K$ to 10 in this paper). Formally, the memorization score $F_m(\overline{x}, G)$ of LLM $G$ on text sequence $\overline{x}$ is computed as follows:

$$F_m(\overline{x}, G) = \frac{1}{|\mathcal{M}_K(\overline{x})|} \sum_{\overline{x}_i \in \mathcal{M}_K(\overline{x})} \log p_G(\overline{x}_i | \overline{x}_{1:i-1}), \tag{1}$$

where $p_G(\overline{x}_i | \overline{x}_{1:i-1})$ denotes the generation probability of token $\overline{x}_i$ by $G$ given its prefix subsequence as context, and set $\mathcal{M}_K(\overline{x})$ includes the $K\%$ of tokens with the lowest probabilities. The higher $F_m$ is, the more likely $\overline{x}$ is memorized by the LLM, *i.e.*, the least memorized content could still been extracted with a high probability. Note that our objective is not to achieve high-precision quantification of memorization. Instead, we aim to statistically demonstrate a valid quantification of memorization across vast samples for comparative analysis.

## 3.3 MEASURING LLM CAPABILITY WITH TRINEVAL

We next present TrinEval, a novel evaluation framework designed to provide a more reliable measure of LLM performance by minimizing the influence of rote memorization.

To understand how LLMs store and manipulate knowledge, Allen-Zhu & Li (2023) created a fictional biography dataset that enumerates various attributes (*e.g.*,, names, jobs, universities) and trained LLMs on this dataset. They employed a linear query-based probing method to uncover correlations between the entity token embeddings and the associated attributes, revealing that where LLMs encode knowledge, *e.g.*, under person names or sequence of the knowledge mention, is crucial for robust mastery of knowledge. This insight leads us to believe that entity tokens, which should ideally store related knowledge, are the target for evaluating an LLM's genuine capability.

However, applying this method to real-world datasets, such as MMLU, presents challenges. Unlike controlled datasets with explicitly defined attributes, real-world data includes a far broader range of possible knowledge. As a result, we cannot enumerate all potential attributes and directly apply linear probing. To this end, we propose TrinEval, a verbal query probing method that reformulates MCQs around a knowledge-centric trinity: knowledge entity, attribute, and context. TrinEval is a pluggable augmentation on any MCQ benchmarks and could expose the genuine capability of LLMs by verifying whether they have correctly encoded knowledge. Generally, we instruct LLMs to extract the triplet according to the prompt and check and refine it if they are not qualified. The

reformulation is completed by a two-round reflection-based prompting method, with the detailed procedure (Alg. 1 in Appendix B) and related prompts available in Appendix B. Here, we explain the elements of the trinity and how to reformulate them.

**Knowledge entity**. We suppose that if an LLM has mastered some knowledge, the key information pertinent to the knowledge should be encoded within a few subject tokens, namely knowledge entity, to support efficient retrieval. By isolating these tokens, TrinEval ensures that only the essential information is considered.

**Attribute**. The attribute acts as a verbal probe to guide the model focusing on the specific feature or property of the knowledge entity being inquired. This mechanism allows TrinEval to isolate and assess the model's understanding of the critical aspects of the questioning subject.

**Context**. In a certain portion of questions, the conditions or background context can significantly influence the solution approach. By explicitly including context in the evaluation process, TrinEval helps the model account for relevant situational details that might otherwise be overlooked, ensuring that the model's answer is based on a comprehensive understanding of the problem.

By extracting the core and necessary question information in this trinity format, the reformulation by TrinEval is knowledge-preserving for the purposes of assessment. In the meanwhile, it completely destructs the original token sequence, effectively reducing the influence of memorization. We will empirically verify these properties through experiments.

Given an MCQ $x = (x_Q, x_O, x_W)$, it first queries a capable reformulation LLM to derive the knowledge entity $x_E$, attribute $x_A$, and context $x_C$ from the original $x$. The LLM is instructed that the triplet should be sufficient for answering the question correctly, without including the answer option itself, ensuring the integrity of the evaluation. The same LLM then assesses whether the triplet contains all necessary information and no redundant information (typically, the rote-memorization), in the meanwhile, yields a rationale $x_L$ as reflection (Shinn et al., 2024; Yao et al., 2022). If it does, the triplet is returned as the re-formulated question. Otherwise, the reformulation model refines the extraction, taking as input $x_E$, $x_A$, $x_C$, and $x_L$, and re-evaluates the updated triplet.

Finally, prompting with the extracted $x_E$, $x_A$, $x_C$, and options $x_O$, we inspect the generation probability of the next ground-truth answer token $x_W$ (*i.e.*, A/B/C/D) as the measurement of capability:

$$F_c(x, G) = p_G(x_W | x_E, x_A, x_C, x_O). \tag{2}$$

As can be seen, $F_c$ metric retains the necessary knowledge-centric information while discarding unnecessary biases, especially the rote memorization of LLMs, which leads to the quantification of genuine capability of LLMs.

## 4 EXPERIMENTS

In this section, we conduct extensive experiments to answer the following questions:

**Q1.** Does TrinEval effectively preserve knowledge to enable accurate knowledge assessment?

**Q2.** To what extent does TrinEval mitigate memorization effects during capability evaluation?

**Q3.** What insights does TrinEval provide into the distinction between rote memorization and genuine capability in LLMs?

### 4.1 EXPERIMENT SETUP

**Models**. We employ API-accessible commercial LLMs for question reformulation in TrinEval, specifically gpt-4o-2024-08-06 (GPT) (OpenAI, 2024) and qwen-max-2024-09-19 (Qwen) (Yang et al., 2024; Team, 2024). For model evaluation, we utilize open-source LLMs to obtain token-level logits. Our experiments focus on three widely adopted models: Llama2-7B (Llama) (Touvron et al., 2023), Mistral-7B-v0.2 (Mistral) (Jiang et al., 2023), and Vicuna-v1.5-7B (Vicuna) (Zheng et al., 2023b). These models are accessed from Hugging Face and implemented with the transformers library, allowing us to compute the log-probabilities of generated tokens for fine-grained analysis. All evaluations are conducted on a single NVIDIA A100 GPU with 40GB of memory. Default generation parameters are used, and greedy decoding is applied throughout to ensure reproducibility.

**Benchmarks**. We evaluate LLMs on the widely used MMLU benchmark (Hendrycks et al., 2020), in light of growing evidence of data contamination (Touvron et al., 2023) in many recent LLMs on this data. MMLU spans 57 subjects across STEM, humanities, social sciences, and others, providing a comprehensive assessment of model capabilities. We find duplicated MCQs across subjects on MMLU, and thus de-duplicate the dataset, resulting in a test set of 14,006 unique questions.

**Evaluation**. With commercial LLMs, we evaluate model performance by extracting predicted answers with regular expressions. For open-source models, we access the output probability of the first generated token (*i.e.*, corresponding to multiple-choice option IDs A/B/C/D) to compute quantitative performance metrics.

## 4.2 Q1. IS TRINEVAL KNOWLEDGE-PRESERVING?

We begin by verify whether TrinEval's reformulation preserves essential knowledge, a prerequisite for reliable knowledge assessment. Specifically, our goal is to ensure that (1) the reformulation does not omit critical information that would cause originally correctly answered questions to be answered incorrectly, and (2) it does not introduce anomalous or unexpected content that could artificially inflate performance.

Following the complete TrinEval reformulation pipeline, we obtain 4,343 qualified MCQs with corresponding knowledge entities, attributes, and contexts using Qwen, and 4,645 MCQs with associ-

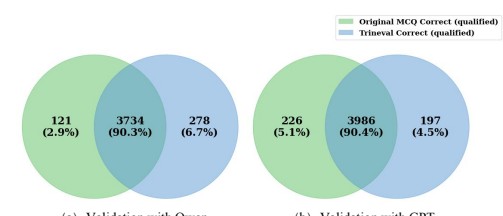

(a) Validation with Qwen    (b) Validation with GPT

Figure 3: Knowledge-preserving validation of TrinEval reformulation. The green, blue, and overlapping regions represent the sets of MCQs correctly answered in the original format, TrinEval format, and both formats, respectively. Best viewed in color.

ated triplets using GPT. We then prompt both Qwen and GPT to answer their respective MCQs in both the original and reformulated triplet-based formats. Example MCQs and the questioning prompt templates in both formats are provided in Table 2 in the Appendix A; results are summarized in Fig. 3. Here, we primarily employ robust API-based LLMs to evaluate the knowledge retention capability of the TrinEval framework. This approach effectively mitigates potential interference from the models' memorization effects, thereby ensuring accurate assessment of their intrinsic knowledge processing abilities and preventing erroneous conclusions that might arise from conflating memorization with genuine comprehension.

Among the qualified MCQs reformulated by Qwen and GPT, 4,133 and 4,409 are answered correctly in at least one of the two formats. Notably, the majority of these are correctly answered in both original and TrinEval formats, accounting for 90.3% with Qwen and 90.4% with GPT, indicating strong consistency in terms of problem-solving across formats. Only 121 questions for Qwen (2.9%) and 226 for GPT (5.1%) are answered correctly exclusively in the original format, suggesting the TrinEval reformulation does not omit essential information. Conversely, 278 questions for Qwen (6.7%) and 197 for GPT (4.5%) are answered correctly only in the reformulated format, which is comparable in general to the numbers for the original format alone. This further suggests that the reformulation does not introduce extra information that might artificially enhance model performance. Collectively, these findings provide strong evidence that TrinEval preserves the core knowledge necessary for answering, satisfying the requirements for reliable capability evaluation.

## 4.3 Q2. CAN TRINEVAL REDUCE MEMORIZATION?

In this subsection, we investigate whether the proposed TrinEval reformulation can suppress unnecessary memorization, thereby isolating and revealing the LLMs' genuine capabilities under various circumstances. Following Bordt et al. (2024), we introduce dataset-specific cues into the questioning prompts and assess to what extent TrinEval mitigates memorization elicited by such prompts.

To evoke memorization, we embed dataset-related metadata into the input prompts, *i.e.*, the dataset name and in-context few-shot examples drawn from the same dataset (see Appendix E). For this set

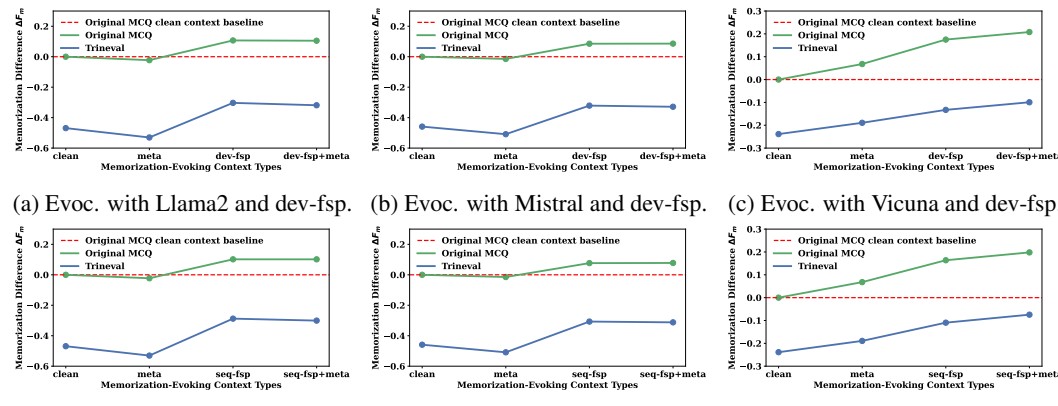

(a) Evoc. with Llama2 and dev-fsp.    (b) Evoc. with Mistral and dev-fsp.    (c) Evoc. with Vicuna and dev-fsp.

(d) Evoc. with Llama2 and seq-fsp.    (e) Evoc. with Mistral and seq-fsp.    (f) Evoc. with Vicuna and seq-fsp.

Figure 4: The results of memorization evocation (evoc.) under various dataset-related context, with green and blue curves referring to the memorization difference $\Delta F_m$ in the original and TrinEval formats, respectively. In the x-axis, 'clean', 'meta', 'dev-fsp', and 'seq-fsp' stand for without dataset-related context, with the name of the dataset, with few-shot prompt from the development set, and with few-shot prompt from the test set ahead of the testing question. These results of $\Delta F_m$ indicate the growing memorization effect given the increasing dataset-related information in general. However, the $\Delta F_m$ by TrinEval under the strongest memory evocation context remains consistently lower than the one in the original format, *e.g.*, the red dashed line.

of experiments, we focus on LLaMA, Mistral, and Vicuna, as their open-source implementations allow access to token-level output probabilities, enabling us to compute the memorization metric $F_m$. Since $F_m$ lacks a defined absolute zero point indicating complete absence of memorization, we use the $F_m$ value obtained from the original MCQ format without any additional context (*i.e.*,, the vanilla MCQ) as a reference baseline. We then visualize the average change in $F_m$ for each memorization-evoking method relative to this baseline.

Specifically, we hypothesize that unintentional data contamination may occur when LLMs are pre-trained on datasets scraped from public sources (*e.g.*,, Hugging Face), where nearby examples from the same dataset may be concatenated during corpus construction. Following the pretraining mechanism of next-token prediction and prior findings from Carlini et al. (2022a), we posit that the inclusion of dataset-similar samples in the prompt may inadvertently trigger memorization. Accordingly, we design a sequence of memorization-evoking perturbations by progressively increasing the contextual cues: (1) providing only the dataset name, (2) adding few-shot examples from the same dataset (*i.e.*,, samples in the development set or ahead examples in the test set of the same subject of MMLU), and (3) combining both types of context. For each MCQ, we compute the change in $F_m$ relative to the reference baseline, capturing the degree to which memorization is elicited by specific context in original or TrinEval formats. Results are illustrated in Fig. 4.

Consistent with the observations by Bordt et al. (2024), the results show that $F_m$ increases as more dataset-specific context is introduced, indicating stronger memorization effects. However, across all three tested LLMs, the $\Delta F_m$ curve for TrinEval remains consistently lower than that of the original MCQ format (*i.e.*, the red dashed line) regardless of various context used. Notably, even under the strongest memorization-evoking setting, TrinEval's absolute $F_m$ still stays below the vanilla baseline. These findings provide compelling evidence that TrinEval significantly mitigates the influence of memorization, effectively disentangling spurious recall from genuine model capability.

## 4.4 Q3. TRINEVAL'S FINDINGS ON LLM MEMORIZATION AND GENUINE CAPABILITY

In this subsection, we aim to explicitly examine the relationship between memorization and genuine problem-solving capability in LLMs, using the metrics $F_m$ (memorization) and $F_c$ (capability). Since commercial API-based models do not expose full vocabulary-level output probabilities, we focus on open-source LLMs for computing these metrics. After computing the $F_m$ and $F_c$ scores for all qualified MCQs in TrinEval format, we divide the values of $F_m$ and $F_c$ into five equal intervals,

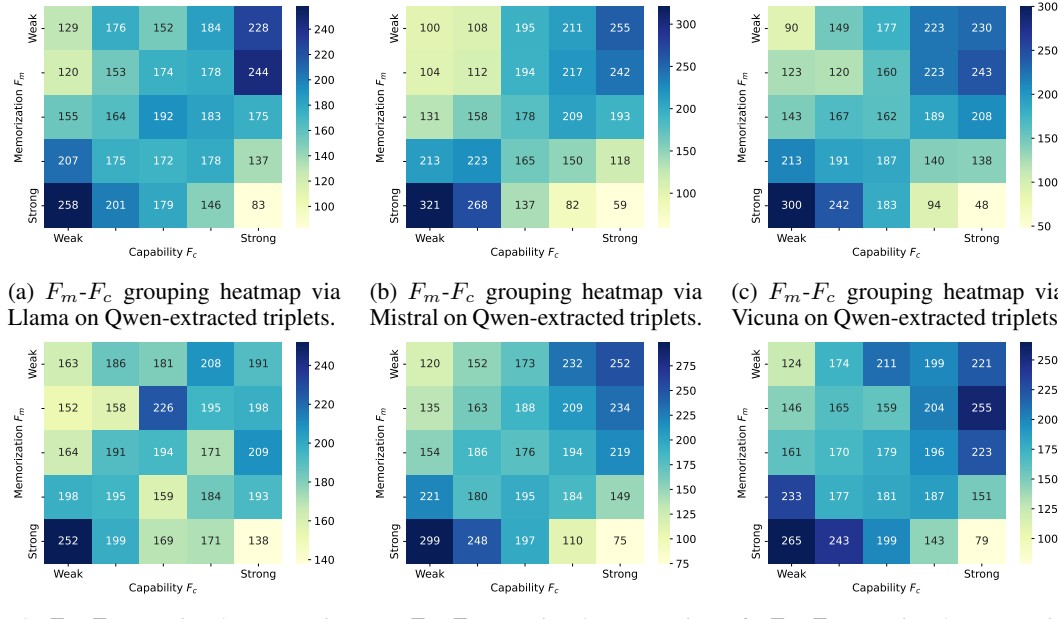

(a) $F_m$-$F_c$ grouping heatmap via Llama on Qwen-extracted triplets.

(b) $F_m$-$F_c$ grouping heatmap via Mistral on Qwen-extracted triplets.

(c) $F_m$-$F_c$ grouping heatmap via Vicuna on Qwen-extracted triplets.

(d) $F_m$-$F_c$ grouping heatmap via Llama on GPT-extracted triplets.

(e) $F_m$-$F_c$ grouping heatmap via Mistral on GPT-extracted triplets.

(f) $F_m$-$F_c$ grouping heatmap via Vicuna on GPT-extracted triplets.

Figure 5: The distribution of MCQs based on memorization metric $F_m$ vs. capability metric $F_c$. According to the values of $F_m$ and $F_c$, we separate the MCQs into 25 groups and visualize the MCQ distribution from weak to strong with heatmaps.

respectively, resulting in 25 distinct groups of MCQ samples based on their joint distribution. We then use heatmaps to visualize the distribution of samples across these groups, thereby revealing the relationship between memorization and capability for each tested model.

As shown in Fig.5, the majority of MCQs cluster in the lower-left and upper-right corners of the heatmap. This pattern is evident and consistent across our tests. For instance, in the results of LLaMA using Qwen-extracted triplets, 38.57% of MCQs fall within the combined lower-left and upper-right $2 \times 2$ grid regions, yielding a Pearson correlation of -0.7755 (p-value $< 0.05$) for $F_m$ vs. $F_c$ of examples within the two square regions; expanding to the $3 \times 3$ regions increases coverage to 74.17%, with a stronger correlation of -0.8124 (p-value $< 0.05$). Similarly, for Mistral on Qwen triplets, 44.90% of MCQs lie in the two $2 \times 2$ corner regions with a correlation of -0.8722 (p-value $< 0.05$), and 80.82% in the two $3 \times 3$ corner regions with a correlation of -0.8794 (p-value $< 0.05$).

We take the MCQs within the lower-left $2 \times 2$ squares in Fig. 5 as the ones memorized by rote and compute the ratio of 20.5% by averaging the percentage of all evaluated LLMs. Similarly, 19.6% is computed as the ratio of MCQs within the upper-right $2 \times 2$ squares with genuine capability. Additional statistics results are reported in Table in Appendix C. These findings consistently indicate a negative correlation between memorization and capability: MCQs with lower memorization scores are more likely to reflect genuine problem-solving capability, whereas high memorization levels correspond to degraded performance, suggesting overfitting or shallow recall.

To interpret these results, we draw an analogy between LLM behavior and the human memory system, which comprises two key components: Short-Term Memory (STM) and Long-Term Memory (LTM) (Shiffrin, 2003). Neurobiological studies reveal that STM relies on transient synaptic protein synthesis with limited temporal persistence and functional scalability. In contrast, LTM is constructed through stabilized neuronal memory traces that constitute an enduring knowledge framework. This neural architecture not only supports STM operations as a cognitive substrate but also enables sophisticated information generalization across diverse contexts. This dichotomy aligns with recent observations in LLMs. As shown by Allen-Zhu & Li (2023) and Ovadia et al. (2023), models trained on corpora with diverse rephrasings exhibit stronger generalization than those trained solely on original formulations. When trained on a single fixed corpus format, LLMs, due to their

strong memorization capacity, tend to activate their STM system to memorize at the token level, capturing surface patterns rather than abstract knowledge. In this sense, **LLMs are potential rote learners**. Reformulations like TrinEval appear to facilitate the encoding of knowledge in a more principled, structured and reusable form, akin to LTM, thereby supporting more generalized and robust evaluation. Further experimental details and results are provided in Appendix F.

To further validate our hypothesis, we analyze the semantic proximity of MCQs within the qualified MMLU dataset by computing their embeddings and measuring the average distance to their closest 1% of samples. Our underlying assumption is that LLMs could better master a knowledge point if it is expressed in various formats in the training corpus. As a result, these knowledge 'rephrases' are also encoded densely via LLMs. By selecting only the top 1% nearest neighbors, we aim to retain the samples that are semantically-correlated in knowledge description. We then compare the mean embedding distances of MCQs located in the lower-left (strong memorization, weak capability) and upper-right (weak memorization, strong capability) $2 \times 2$ regions of the heatmap in Fig. 5. The corresponding results are presented in Fig. 6. Interestingly, we observe that the average embedding distance among the Genuine Capability Learning MCQs is much lower than (nearly half) that of the Rote Memorization MCQs. This suggests that memorized MCQs are more sparsely distributed in the embedding space, while non-memorized, capability-driven MCQs tend to form tighter semantic clusters. This observation also aligns well with the cognitive analogy of STM and LTM: rote memorization leads to fragmented, context-specific encoding, whereas genuine capability emerges from structured, reusable representations. The results of alternative thresholds are shown in Table in Appendix. Though it is well believed that memorization may lead to better but cheating performance of LLMs, we find that the more LLMs memorize, the worse they are at solving problems.

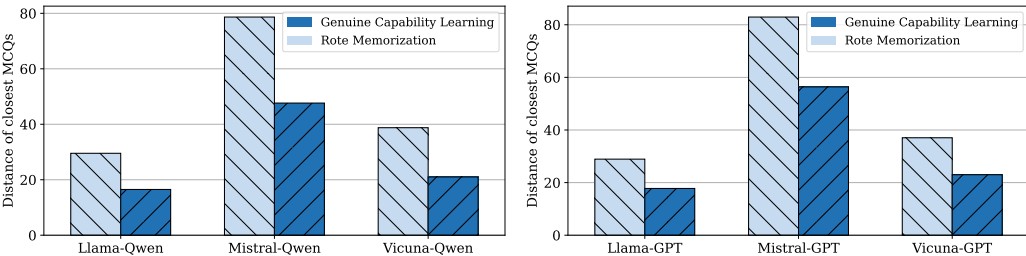

Figure 6: Averaged distance of each MCQs between the closest $1\%$ MCQs' embeddings. 'Rote Memorization' refers to MCQ within lower left $2 \times 2$ squares that typically exhibits high memorization metric $F_m$ and low capability $F_c$ while the 'Genuine Capability Learning' stands for MCQ lies within the upper right $2 \times 2$ squares with low $F_m$ but high $F_c$. Further results are shown in Appendix F.

## 5 CONCLUSION

This study provided a novel perspective on benchmark contamination in LLM evaluation, reframing it as an inherent aspect of learning. This perspective led us to explore the relationship between memorization and genuine capability in LLMs. Through our empirical investigation, we observed a surprising result: LLMs performed worse on memorized MCQs compared to those not, suggesting that superficial memorization may undermine problem-solving ability rather than enhance it. This finding also implies the existence of two distinct learning paradigms in LLMs: rote memorization and genuine capability learning.

To disentangle them, we proposed TrinEval, an evaluation method reformulating MCQs into a knowledge-centric trinity and separating the influence of memorization from genuine knowledge application. Experiments validated both knowledge-preserving and memorization-reducing properties of TrinEval. Based on that, TrinEval reveals the in-robustness of LLMs' knowledge learning, *e.g.*, popular open-source LLMs memorize 20.5% of knowledge points by rote in MMLU. We also discussed the generalizability of our approach on other forms of questions (*e.g.*, open-ended questions) in Appendix G. As such, we believe this work lays the groundwork for future studies on improving LLM knowledge robustness and more thorough evaluation.

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

## A  DETAILS OF THE EXTRACTING PROMPTS AND THE EXTRACTED (MEMORIZED) MCQS

In this section, we introduce the details of the processed dataset and the prompts for extraction. MCQs from some subjects contain similar or identical options[3]. With the provided 5-shot prompt, options of MCQs from these subjects can be easily extracted, leading to a high False-positive ratio. In order to avoid the influence of the few-shot prompt on the option extraction, we eliminate MCQs in which any of the options have appeared twice in the dataset. After deduplication, we obtain 14,006 MCQs for evaluation. The extraction prompt and the detailed statistics are shown in the following text and Tab. 1.

## B  DETAILS OF TRINEVAL

In this section, we introduce the details of the proposed TrinEval. The pseudo-code is shown in the Alg. 1. The prompts used are also shown below. Note that the potential data leakage is often caused by the data crawled on the Huggingface dataset site. Thus, we also provide the original text of MCQs in the format on the Huggingface dataset site to mimic the data contamination with in-context learning.

---

[3]E.g., the options of MCQs in the subject, moral_scenarios, are all identical ('Wrong, Wrong', 'Wrong, Not wrong', 'Not wrong, Wrong' and 'Not wrong, Not wrong').

| Prompt template for extraction: | Prompt template for pre-investigation on LLM Memorization w.r.t. Capability: |
|---|---|
| *You are an expert of multiple choice questions of MMLU dataset. The following are multiple-choice questions (with answers) about [subject].*

*[examples]*

*[question]*
*Options:*
*A.* | *You are an expert of multiple choice questions of MMLU dataset. The following are multiple choice questions (with answers) about [subject].*

*[examples]*

*[question]*
*Options:*
*A. [content for option A]*
*B. [content for option B]*
*C. [content for option C]*
*D. [content for option D]*

*Answer:* |

| Model | Subset | Simple | Pro | MMLU |
|---|---|---|---|---|
| | memorized | 912 | 70 | 982 |
| Llama | non-mem. | 6,548 | 6,476 | 13,024 |
| | all | 7,460 | 6,546 | 14,006 |
| | memorized | 879 | 36 | 915 |
| Mistral | non-mem. | 6,581 | 6,510 | 13,091 |
| | all | 7,460 | 6,546 | 14,006 |
| | memorized | 893 | 16 | 909 |
| Vicuna | non-mem. | 6,567 | 6,530 | 13,097 |
| | all | 7,460 | 6,546 | 14,006 |

Table 1: Statistics of memorized and non-memorized questions by Llama2-7B, Mistral-7B-v0.2, and Vicuna-v1.5-7B in MMLU.

## C    DETAILED RESULTS OF MEMORIZATION V.S. CAPABILITY

In this section, we exhibit the detailed results of the Q3. What does TrinEval reveal about the memorization v.s. the capability of LLMs. We reveal the ratio of MCQs within the upper right and lower left $2 \times 2$ and $3 \times 3$ squares as well as the Pearson correlations between the $F_m$ and $F_c$ of these MCQs. Our analysis reveals a tendency towards a negative correlation between the capabilities and memorization of LLMs shown in the Tab. 3. All person correlation values are computed with p-values at the 0.05 level.

Further, inspired by the Precision-Recall Curve, we take each unique $F_m$ of the qualified MCQs as the threshold to separate them as the Memorized and Capable MCQs. For each separation, we compute the probability of whether the $F_c$ of a randomly selected Capable MCQ exceeds the $F_c$ of a randomly selected Memorized MCQ and plot them as the blue curve. We also compute the T-test p-value between the $F_c$s of the Memorized MCQs and Capable MCQs as the green curve. The results are shown in Fig. 7. For the second row, we filter out the MCQs within the upper left and lower right $2 \times 2$ squares. From the figure, we observe that over a relatively long segment in the middle of the x-axis threshold range, the probability remains at a comparatively high value, while the p-value stays below 0.05. From this, we can conclude that $F_m$ can distinguish between MCQs with high $F_c$ and those with low $F_c$ with a negative correlation at a high confidence level. This further supports that LLMs are potential rote learners, the more the LLMs memorize, the more poorly they perform.

---

**Algorithm 1** MCQ reformulation by TrinEval

---

**Input:** Question $x_Q$, options $x_O$, and answer $x_W$ of an MCQ.

**Output:** Reformulated question $x_Q^R$.

1: Preliminarily extract knowledge entity $x_E$, attribute $x_A$, and context $x_C$ based on $x_Q$, $x_O$ and $x_W$;

2: Initialize $X_Q^R = x_E, x_A, x_C$;

3: Validate the adequacy and necessity of the $x_Q^R$ and give reasons $x_L$;

4: **if** $x_Q^R$ matches the requirement **then**

5:     Return $x_Q^R$;

6: **else**

7:     Re-extract $x'_E$, $x'_A$, and $x'_C$ by reflecting with $x_E, x_A, x_C$ and $x_L$;

8:     Update $x_Q^R = x'_E, x'_A, x'_C$;

9:     Validate the adequacy and necessity of the $x_Q^R$ and give reasons $x_L$;

10:     **if** $x_Q^R$ matches the requirement **then**

11:       Return $x_Q^R$;

12:     **else**

13:       Discard the MCQ, return $None$;

14:     **end if**

15: **end if**

---

(a) Probability and p-value with Llama2 based on GPT.

(b) Probability and p-value with Mistral based on GPT.

(c) Probability and p-value with Vicuna based on GPT.

(d) Probability and p-value with Llama2 based on GPT (21.59 % MCQs filtered).

(e) Probability and p-value with Mistral based on GPT (15.95 % MCQs filtered).

(f) Probability and p-value with Vicuna based on GPT (17.59 % MCQs filtered).

Figure 7: The over-performing probability curve and p-value curve with different $F_m$ thresholds. In this figure, we take each unique $F_m$ as the threshold to separate the qualified MCQs into the Memorized and non-Memorized MCQs. We compute the probability of a randomly selected non-Memorized MCQ's $F_c$ exceeding a randomly selected Memorized MCQ's $F_c$ under each threshold as the blue curve, and the green curve is the p-value of the T-test between the $F_c$s of the non-Memorized MCQs and the Memorized MCQs.

## D   HUMAN ANNOTATION

As there is potential risk for the LLM-based Knowledge Preserving evaluation in TrinEval procedure (line 4 and line 10 in 1) that the API-based LLMs might still be able to answer the MCQs without sufficient knowledge since this is still prompting LLMs who have been trained on these datasets and "know" the original content, human annotation is also applied. The annotation of each MCQ encompasses three subtasks: (1) answering the question in the re-organized form, (2) answering the question in the original form, and (3) verifying if the reformulation is Knowledge Preserving or not.

For efficient annotation, we implemented a stratified sampling procedure by selecting one MCQ per subject from all 56 MMLU subjects (as a temporary compromise for limited time, which will be expanded later) under both Qwen and GPT reorganization paradigms. This yielded 112 representative questions (2 systems × 56 subjects) for evaluation. Three human annotators independently performed dual-form assessments through: (1) Direct question answering with the reformed format first and the original format; (2) Knowledge Preservation (K.P.) scoring across two dimensions: i. Knowledge adequacy (sufficiency for accurate response), ii. removal of redundant content using a 5-point scale (1=unsatisfactory; 2=major information are missed or unnecessary information is incorporated, but part is still acceptable, 3=need to take some time to understand, but can still solve the MCQ, 4=an element properly belonging to one triplet component appears in another, but does not impact MCQ solving; 5=optimal). We use a continuous rather than binary metric to mitigate the cognitive difference of the threshold between the annotators. Inter-rater reliability was ensured through consensus-building discussions prior to formal annotation. Final scores were aggregated using mean values to further mitigate individual annotator bias. The results are shown in Tab. 4 below.

Notably, our analysis reveals that over 95% of correctly answered MCQs maintained consistency across both original and paraphrased formats. Furthermore, human annotators rated our paraphrased questions mean K.P. scores exceeding 4.0 (on a 5-point scale), which means that the reformulated MCQs only somehow influence the readability of humans but do not impact the solvability of the original format. This provides empirical validation that our proposed TrinEval methodology effectively preserves necessary knowledge elements from original MCQ formulations, while the influence of the LLM memorization during the evaluation is rather limited.

Experimental results under human annotation also reveal that Qwen underperforms GPT in key metrics, particularly in processing long-context texts where it occasionally omits background information (evidenced by excessive "N/A" assignments in the Context fields). This capability gap is further reflected in MCQ annotations: there are only MCQs that are merely correctly answered with the original format except for the correct MCQs with both formats.

## E  ELABORATION ON DATASET-RELATED INFORMATION

We suppose that unintentional data contamination arises from crawling dataset pages (e.g., Hugging Face) during the compilation of LLM pretraining datasets. When researchers are organizing the pretraining corpus, one or more neighboring original data samples would be truncated and concatenated sequentially into a pretraining sample. Thus, according to Carlini's theory Carlini et al. (2022a) and the next-token-prediction pretraining, we believe that offering samples within the same dataset would affect memorization evocation.

Besides, the previous study Bordt et al. (2024) also applies a similar method, the "Header Completion Test", for tabular data memorization detection. By offering the heading rows within a CSV file, they also find that providing the preceding data samples can help to detect the memorized dataset by LLMs, which also practically proves that offering samples within the same dataset would affect the memorization evocation.

Following this, similar to the dataset name, we take the samples within the same dataset as the few-shot prompt in order to find out if the TrinEval re-organization method can avoid such memorization evocation phenomenon. Still, in Fig. 4, we can see that the blue curves remain below the green ones for the "def-fsp" setting, which proves that TrinEval can restrain the memorization evocation.

## F  EMBEDDING DISTANCE OF MEMORIZED AND NON-MEMORIZED MCQS

As there are 57 different subjects within the MMLU dataset, we believe unrelated sequences would lead to increased embedding distance even though they are among the mastered knowledge points. Here, we try to filter out the unrelated samples for each sample and thus filter the closest samples at the $1e-2$ level in order to make sure there are not too many unrelated samples incorporated.

To highlight this result more prominently, we employed a relatively stringent data filtering strategy in the paper and made it 1% of the closest samples in Section 4.4. In the following version, we will add this clarification part in the paper. Here in order to provide a more robust result, we also

present results obtained under more lenient data filtering criteria, such as thresholds of 3% and 5%. The results are shown in Tab. 5 (RM stands for rote memorization, and GCL stands for genuine capability learning).

We can see that, as we said above, the more samples we incorporated, the higher the average distance of the closest embeddings grows. Still, though we increase the filtering threshold and the distance gap between the RM and the GCL is narrowing, we can still find that the distance between the closest rote memorization MCQ embeddings is more than the distance between the closest genuine capability learning MCQ embeddings. This proves that the reported results are robust and solid.

## G    LIMITATIONS

Our limitations are mainly three points. First, though we mainly use the MCQs as the tested benchmarks, We believe that rote memorization and genuine capability learning are among the most essential for understanding LLM learning. In the current study, we choose MCQs as the testbed due to its wide adoption in LLM evaluation and easily verifiable response correctness. This has led to several innovations, such as the attempt to evaluate LLMs in the presence of contamination, the interintuitive trend of model performance under different memorization conditions, and the quantification of rote memorization and genuine capability learning.

On the other hand, our disentangling point of view and the concrete methodology should also apply to problems in forms other than MCQs, e.g., open-ended QA problems. Note that it could be possible to recruit human experts to write responses to questions from these classic evaluation benchmarks, says TruthfulQA, SuperGELU, Arena-Hard, and AlpacaEval 2.0. Using these responses during SFT could also result in inflated performance. It would be very interesting to compare the performance before and after question reformulation that reduces rote memorization to intentionally injected responses. We will discuss such potential applications in the next version. Thank you again for your comment.

Second, though our proposed TrinEval retrains the problem-solving ability of the LLMs and obtains stronger robustness, it is not a dynamical re-organizing method that can still be leaked and pre-experienced during training. On the one hand, we appeal to the LLM developers not to use this re-organizing method as part of the training corpus. On the other hand, future works will be focused on developing dynamic evaluation method (Zhu et al., 2023; 2024).

Finally, we did not give a clear exploration on how and why the more LLMs memorize, the less the capability of the LLMs obtains. In future work, we will also look into the mechanism of the training and structure of LLMs for a thorough study of the phenomenon.

**Prompt template for triplet extraction:**

*You are an expert of Knowledge Keyword extraction. Analyze and summarize the Question based on the given Fact corpus and extract the Knowledge Keyword, the Attribute and the Context (if necessary) within the Question.*

*Given a Fact corpus, a Question about the Fact corpus, and the Answer to the Question, analyze the Question corpus as well as the given Answer. Applying the provided steps, extract the Knowledge Keyword, the Attribute of the Knowledge Keyword and the necessary Context to obtain the key information of the Question, ensuring they are sufficient for answering the given Question and obtaining the given Answer.*

*# Steps*

*1. **Review the Fact corpus:** Read through the entire Fact corpus to understand the context.*

*2. **Identify the Question:** Focus on the given Question to capture which part of the Fact corpus it is asking about.*

*3. **Understand the Answer to the Question:** Compare the given Answer and the identified questioned part within the Fact corpus and understand why this answer was chosen.*

*4. **Write Step-by-Step Reasoning:***
*- Identify the asked Knowledge Keyword in the Question that is the subject of the most information in the Fact corpus and the asked Question is about the information among.*
*- Determine the asked Attribute of the Knowledge Keyword in the Question, which can be used to infer the given Answer.*
*- Review the identified Knowledge Keyword and Attribute to confirm that only these two parts can be used to obtain the given Answer to the given Question. If not, extract all the necessary Context from the Question that makes it enough to obtain the given Answer to the given Question.*

*5. **Determine Outcome:** Based on the reasoning, conclude and extract the Knowledge Keyword, the Attribute and the Context (if necessary) of the Question according to the Question corpus.*

*# Output Format*

*Provide the outcome in the following format:*

*- **Step-by-Step Reasoning:** [Detailed reasoning here]*
*- **Knowledge Keyword:** [Extracted Knowledge Keyword here]*
*- **Attribute:** [Extracted Attribute of the Knowledge Keyword here]*
*- **Context:** [Extracted Context within the Question to make up for the Knowledge Keyword and the Attribute here if necessary]*

*# Examples*

*[examples]*

*# Notes*

*- Strictly follow the format of the examples and give Knowledge Keywords, the Attribute and the Context (if necessary) anyway.*
*- The extracted Knowledge Keyword, Attribute and Context (if necessary) should be the original text within the Question and should not incorporate any phrases that cannot be exactly matched in the Question.*
*- Never include any information from the options of the multiple choice question, especially the content of the answer option.*
*- The extracted Knowledge Keyword, Attribute and Context (if necessary) should include all the necessary information only within the Question Corpus for answering the Question and obtaining the given Answer.*

***Fact:** [question]   [option content list]   [subject]   [answer option index][answer option ID]*

***Question:** [question]*

***Answer:** [content of the answer option]*

---

**Prompt template for triplet validation & reflection:**

*You are an expert of [subject] and an advanced reasoning agent that can determine whether the given Knowledge Keyword, Attribute of the Knowledge Keyword and the Context present most of the necessary information of the Question for obtaining the given Answer. Suppose you have sufficient background knowledge about subj. Consider the given Knowledge Keyword, Attribute and the Context, then determine whether the given Answer can be directly obtained from them even without the Question.*

*# Steps*

*1. \*\*Check the Semantic completeness:\*\* Suppose you have sufficient background knowledge about [subject], and you can solve the given Question and obtain the given Answer. Read through the given Knowledge Keyword, Attribute, Context and the given Question. Check if the given Knowledge Keyword, Attribute, Context are the original text within the Question and contain the necessary queried information the Question itself provided (ignore the information the Question did not provided). If not so, check if the missed information is indeed incorporated in the Question (which is not acceptable, but if not, it is acceptable). Point out the information that is within the Question but they have missed. Then in a few sentences, diagnose the possible reason for failure or the phrasing discrepancy, and devise new, concise, high-level improvement suggestions to avoid the same failure.*

*2. \*\*Check the Answer relevance:\*\* Suppose you have sufficient background knowledge about subj, and you can solve the given Question and obtain the given Answer. Read through the given Knowledge Keyword, Attribute, Context and the given Question. Read through the given Knowledge Keyword, Attribute, Context and the given Answer. Check if the Answer can be directly inferred with the given Knowledge Keyword, Attribute and the Context without seeing the Question. If not so, check if the missed information is indeed incorporated in the Question (which is not acceptable, but if not, it is acceptable). Point out the information that is within the Question but they have missed. Then in a few sentences, diagnose the possible reason for failure or the phrasing discrepancy, and devise new, concise, high-level improvement suggestions to avoid the same failure.*

*3. \*\*Check the Semantic Redundancy:\*\* Read through the given Knowledge Keyword, Attribute, Context, the given Question and the given corresponding Answer. Check if the Answer can be directly matched within the given Knowledge Keyword, Attribute and the Context. Check if there are any unnecessary information within the given Knowledge Keyword, Attribute and the Context for obtaining the given Answer to the Question. If not so, point out what is redundant. Then in a few sentences, diagnose the possible reason for failure or the phrasing discrepancy, and devise new, concise, high-level improvement suggestions to avoid the same failure.*

*# Output Format*

*Provide the outcome in the following format:*

*- \*\*Step-by-Step Reasoning:\*\* [Detailed reasoning here]*
*- \*\*Verdict for the given Knowledge Keyword, Attribute and Context:\*\* [Single verdict (Yes/No) here for whether the given Knowledge Keyword, Attribute and Context contain most of the asked information of the Question, can be used to infer the given Answer with only them without the whole Question, and do not contain redundant information for obtaining the given Answer.]*

*# Notes*

*- Do not deviate from the specified format. Do not generate anything else after the Verdict (only Yes/No) for the given Knowledge Keyword, Attribute and Context.*
*- Suppose you have sufficient background knowledge about subj, and you can solve the given Question and obtain the given Answer. For Semantic completeness and Answer relevance, it is acceptable to miss information that is also not incorporated in the Question.*
*- Provide a detailed explanation following the given steps before arriving at the verdict (Yes/No). Provide a final verdict (only Yes/No) in order at the end in the given format.*

*- \*\*Question:\*\* [question]*
*- \*\*Answer:\*\* [answer]*

*- \*\*Knowledge Keyword:\*\* [extracted knowledge entity]*
*- \*\*Attribute:\*\* [extracted attribute]*
*- \*\*Context:\*\* [extracted context]*

---

972

**Prompt template for the second round triplet extraction:**

973

974 *You are an advanced reasoning agent that can improve through self-reflection and an expert of Knowledge Keyword extraction. Analyze and summarize the Question based on the given Fact corpus and extract the Knowledge Keyword, the Attribute and the Context (if necessary) within the Question.*

975

976

977 *Given a Fact corpus, a Question about the Fact corpus, and the Answer to the Question, analyze the Question corpus as well as the given Answer. Applying the provided steps, extract the Knowledge Keyword, the Attribute of the Knowledge Keyword and the necessary Context to rephrase the Question, ensuring they are sufficient for answering the given Question and obtaining the given Answer.*

978

979 *# Steps*

980

981 *1. **Review the Fact corpus:** Read through the entire Fact corpus to understand the context.*

982 *2. **Identify the Question:** Focus on the given Question to capture which part of the Fact corpus it is asking about.*

983

984 *3. **Understand the Answer to the Question:** Compare the given Answer and the identified questioned part within the Fact corpus and understand why this answer was chosen.*

985

986 *4. **Write Step-by-Step Reasoning:***
*- Identify the asked Knowledge Keyword in the Question that is the subject of the most information in the Fact corpus and the asked Question is about the information among.*
*- Determine the asked Attribute of the Knowledge Keyword in the Question, which can be used to infer the given Answer.*

987

988 *- Review the identified Knowledge Keyword and Attribute to confirm that only these two parts can be used to obtain the given Answer to the given Question. If not, extract all the necessary Context from the Question that makes it enough to obtain the given Answer to the given Question.*

989

990

991 *5. **Determine Outcome:** Based on the reasoning, conclude and extract the Knowledge Keyword, the Attribute and the Context (if necessary) of the Question according to the Question corpus.*

992

993 *# Output Format*

994 *Provide the outcome in the following format:*

995

996 *- **Step-by-Step Reasoning:** [Detailed reasoning here]*
*- **Knowledge Keyword:** [Extracted Knowledge Keyword here]*
*- **Attribute:** [Extracted Attribute of the Knowledge Keyword here]*

997 *- **Context:** [Extracted Context within the Question to make up for the Knowledge Keyword and the Attribute here if necessary]*

998

999 *# Examples*

1000 *[examples]*

1001

1002 *You will be given a previous trial. You were unsuccessful in extracting the Knowledge Keyword, Attribute and the necessary that meet the requirements in the previous trial. Given the Reflection below, improve the process. The process is as follows:*

1003

1004 *# Previous returns:*

1005 *- **Fact:** [question]   [option content list]   [subject]   [answer option index][answer option ID]*

1006 *- **Question:** [question]*

1007

1008 *- **Answer:** [answer option content]*

1009 *- **Knowledge Keyword:** [extracted knowledge entity of the last trial]*

1010 *- **Attribute:** [attribute of the last trial]*

1011

1012 *- **Context:** [context of the last trial]*

1013 *- **Reflection:***
*[rational of the last trial]*

1014

1015 *# Notes*

1016 *- Consider the Reflection given above. Improve the extraction of Knowledge Keyword, Attribute and Context (if necessary).*
*- Strictly follow the format of the examples and give Knowledge Keywords, the Attribute and the Context (if necessary) anyway.*

1017 *- The extracted Knowledge Keyword should be phrases within the Question and should not incorporate any information of the Fact corpus*

1018 *or the given Answer that is not mentioned in the Question.*

1019 *- The extracted Attribute and Context (if necessary) should only include information from the Question corpus. Never include information from the options of the multiple choice question, especially the content of the answer option.*

1020 *- The extracted Knowledge Keyword, Attribute and Context (if necessary) should include all the necessary information only within the Question Corpus for answering the Question and obtaining the given Answer.*

1021

1022 ***Fact:** [question]   [option content list]   [subject]   [answer option index][answer option ID]*

1023 ***Question:** [question]*

1024

1025 ***Answer:** [content of the answer option]*

| Original MCQ | Original MCQ Example |
|---|---|
| *You are an expert on multiple choice questions of [subject]. Analyze the given question and the given options. Determine the correct answer option to the question.*

*Given a Question and the potential Answer options to the Question, analyze the Question as well as the given options. Generate the option ID of the correct option (answer).*

*- \*\*Question:\*\**
*[question]*

*- \*\*Options:\*\**
*A. [option A]*
*B. [option B]*
*C. [option C]*
*D. [option D]* | *You are an expert on multiple choice questions of high school computer science. Analyze the given question and the given options. Determine the correct answer option to the question.*

*Given a Question and the potential Answer options to the Question, analyze the Question as well as the given options. Generate the option ID of the correct option (answer).*

*- \*\*Question:\*\**
*Which of the following is usually NOT represented in a subroutine's activation record frame for a stack-based programming language?*

*- \*\*Options:\*\**
*A. Values of local variables*
*B. A heap area*
*C. The return address*
*D. Stack pointer for the calling activation record* |
| **TrinEval MCQ** | **TrinEval MCQ Example** |
| *You are an expert on multiple choice questions of [subject]. Analyze the given Knowledge Entity, Attribute of the Knowledge Entity, the Context of a question, and the given options to the question. Determine the correct answer option to the question.*

*The Knowledge Entity is the questioned subject of the question. The Attribute is the questioned attribute of the Knowledge Entity, and the Context is the necessary context information for answering the question. Given a set of Knowledge Entity, Attribute, and Context (which three are extracted as the key information from a question), and the potential Answer options to the Question, analyze the given Knowledge Entity, Attribute, Context as well as the options. Generate the option ID of the correct option (answer).*

*- \*\*Knowledge Entity:\*\**
*[knwoledge entity]*

*- \*\*Attribute:\*\**
*[attribute]*

*- \*\*Context:\*\**
*[context]*

*- \*\*Options:\*\**
*A. [option A]*
*B. [option B]*
*C. [option C]*
*D. [option D]* | *You are an expert on multiple choice questions of high school computer science. Analyze the given Knowledge Entity, Attribute of the Knowledge Entity, the Context of a question, and the given options to the question. Determine the correct answer option to the question.*

*The Knowledge Entity is the questioned subject of the question. The Attribute is the questioned attribute of the Knowledge Entity, and the Context is the necessary context information for answering the question. Given a set of Knowledge Entity, Attribute, and Context (which three are extracted as the key information from a question), and the potential Answer options to the Question, analyze the given Knowledge Entity, Attribute, Context as well as the options. Generate the option ID of the correct option (answer).*

*- \*\*Knowledge Entity:\*\**
*subroutine's activation record frame*

*- \*\*Attribute:\*\**
*usually NOT represented*

*- \*\*Context:\*\**
*for a stack-based programming language*

*- \*\*Options:\*\**
*A. Values of local variables*
*B. A heap area*
*C. The return address*
*D. Stack pointer for the calling activation record* |

Table 2: Template and an example of the Original MCQ template and the TrinEval MCQ template. [·] refers to the blank that should be filled according to the content of each MCQ.

| LLMs | Dataset | $2 \times 2$ squares | | $3 \times 3$ squres | |
|------|---------|----------|---------------------|----------|---------------------|
| | | Ratio (%) | Pearson correlation | Ratio (%) | Pearson correlation |
| Llama2-Qwen | All | 38.57 | -0.7755 | 74.17 | -0.8124 |
| | Simple | 37.07 | -0.7784 | 72.63 | -0.8121 |
| | Pro | 38.66 | -0.783 | 74.51 | -0.8109 |
| Llama2-GPT | All | 35.22 | -0.7835 | 71.04 | -0.7924 |
| | Simple | 33.9 | -0.777 | 69.62 | -0.7919 |
| | Pro | 35.45 | -0.7916 | 71.54 | -0.7881 |
| Mistral-Qwen | All | 44.9 | -0.8722 | 80.82 | -0.8794 |
| | Simple | 38.47 | -0.8494 | 74.04 | -0.8271 |
| | Pro | 44.32 | -0.8045 | 80.08 | -0.8682 |
| Mistral-GPT | All | 40.37 | -0.8042 | 76.58 | -0.8736 |
| | Simple | 35.51 | -0.8297 | 72.27 | -0.8664 |
| | Pro | 38.52 | -0.7103 | 74.91 | -0.7969 |
| Vicuna-Qwen | All | 42.94 | -0.8771 | 79.23 | -0.8365 |
| | Simple | 37.86 | -0.758 | 73.85 | -0.7168 |
| | Pro | 42.01 | -0.8609 | 77.86 | -0.886 |
| Vicuna-GPT | All | 38.69 | -0.8621 | 74.83 | -0.8672 |
| | Simple | 34.77 | -0.8096 | 70.71 | -0.7775 |
| | Pro | 37.37 | -0.7794 | 73.98 | -0.8728 |

Table 3: The ratio and the Pearson-correlation between the $F_c$ and $F_m$ of the MCQs within the upper right and lower left $2 \times 2$ and $3 \times 3$ squares. For LLMs, 'Llama2-Qwen' refers that the $F_c$ and $F_m$ are calculated with Llama2 based on the Qwen-extracted triplet, and similarly hereinafter. For the Dataset column, 'All' stands for all the qualified MCQs after the triplet extraction, 'Pro' refers to the qualified MCQs that are the members of the mmlupro dataset while 'Simple' refers to the rest of the MCQs that are relatively easier.

| Model | TrinEval correct only | both correct | original correct only | K.P. score |
|-------|------------------------|--------------|------------------------|------------|
| Qwen | 0.0% | 96.296% | 3.704% | 4.101 |
| GPT | 0.0% | 96.667% | 3.333% | 4.369 |

Table 4: Result of Human Annotation on LLM-based knowledge preserving (K.P.) evaluation in TrinEval

| Threshold | 1% | | 3% | | 5% | |
|-----------|-----|-----|-----|-----|-----|-----|
| Subset | RM | GCL | RM | GCL | RM | GCL |
| Llama-Qwen | 29.538 | 16.501 | 32.718 | 17.281 | 33.790 | 18.060 |
| Llama-GPT | 28.925 | 17.777 | 31.327 | 18.275 | 32.315 | 19.068 |
| Mistral-Qwen | 78.663 | 47.648 | 86.003 | 52.995 | 89.598 | 55.525 |
| Mistral-GPT | 82.932 | 56.375 | 90.597 | 62.112 | 94.384 | 64.820 |
| Vicuna-Qwen | 38.761 | 21.006 | 41.396 | 23.189 | 42.490 | 24.185 |
| Vicuna-GPT | 37.037 | 22.996 | 41.396 | 23.189 | 42.490 | 24.185 |

Table 5: Result of averaged embedding distance of the closest MCQs under different thresholds

