# OpenReview forum: "Large Language Models Could Be Rote Learners"
_ICLR.cc/2026/Conference — Submitted to ICLR 2026_

### Official Review · Reviewer_rvEY · 2025-10-30

**Soundness:** 3
**Presentation:** 3
**Contribution:** 2
**Rating:** 4
**Confidence:** 4

**Summary:**

This paper investigates "benchmark contamination" in LLMs, distinguishing between genuinely learned capabilities and simple "rote memorization."
The authors find that LLMs often perform worse on perfectly memorized questions from benchmarks like MMLU compared to non-memorized ones. This suggests superficial memorization doesn't equal understanding.
To address this, they propose TrinEval. This framework reformulates multiple-choice questions into a "knowledge entity, attribute, and context" trinity. This breaks the exact phrasing models might have memorized while keeping the core problem intact.
Experiments show TrinEval effectively isolates genuine capability. Results indicate open-source LLMs rote-memorize about 20.5% of MMLU knowledge points, while truly mastering only 19.6%.

**Strengths:**

- Novel Perspective: Instead of just trying to remove contaminated data, it studies how contamination (memorization) actually affects performance, and discovers a counterintuitive finding thatmemorized questions often have lower accuracy.
- Practical Solution (TrinEval): The proposed reformulation method provides a concrete way to evaluate models more fairly, even if they have seen the test data.

**Weaknesses:**

- Limited Scope: The study focuses almost entirely on multiple-choice questions in MMLU. It's unclear how well this applies to open-ended or reasoning tasks.
- Dependence on LLMs for Reformulation: The TrinEval framework relies on LLMs to perform the question reformulation. This raises questions about whether the reformulation process itself introduces any biases from the model used.

**Questions:**

- The core finding that LLMs perform worse on memorized MCQs is fascinating. Do you have a hypothesis as to why this occurs?
- Have you considered applying the TrinEval methodology to other types of evaluation benchmarks beyond MCQs, such as those for mathematical reasoning (GSM8K, MATH) or code generation (HumanEval, LCB)? How might the "knowledge-centric trinity" be adapted for these different formats?
- The analogy to human STM and LTM is insightful. Do your findings suggest that specific training strategies or architectural changes could encourage LLMs to develop more LTM-like knowledge representations and rely less on STM-like rote memorization?

---

> ### Author Response · Authors · 2025-11-26
>
> Dear Reviewer,
>
> Thank you for your thoughtful and constructive comments, which have helped us clarify and strengthen key aspects of our work. In response to your feedback, we have further justified our choice of multiple-choice questions (MCQs) as a controlled experimental setting and explained how our approach can be conceptually extended to open-ended tasks. We have also addressed potential concerns regarding LLM-based reformulation by providing empirical and human-annotation evidence supporting the robustness and limited bias of our method. Additionally, we elaborated on the cognitive and empirical basis for our core finding—that LLMs perform worse on memorized MCQs—and discussed how training strategies could encourage more robust, LTM-like knowledge representations. We also outlined potential adaptations of the TrinEval framework to other benchmarks such as mathematical reasoning and code generation. Below, we provide detailed responses to each of your comments.
>
> **W1. Limited Scope: The study focuses almost entirely on multiple-choice questions in MMLU. It's unclear how well this applies to open-ended or reasoning tasks.**
>
> Thank you for your thoughtful comment. Our study primarily uses multiple-choice questions (MCQs) from the MMLU benchmark. We are also working on the GSM8K dataset. In preliminary experiments, we found that 1282/1319 MCQs qualified after Qwen3-Max-based TrinEval reformulation, while 1192/1319 qualified after DeepSeek-R1-based reformulation. Due to the lack of an objective and fair metric for evaluating LLM responses to open-ended questions, we have temporarily refrained from conducting the Memorization-Capability heatmap analysis. We plan to extend our work in the future once more robust and well-recognized evaluation methods become available. Also, we view open-ended questions essentially as MCQs without provided answer choices. As shown in Appendix Table 2 (Page 20), when the candidate options are removed, the original question can be reformulated into a triplet (knowledge entity, context, attribute), making our method applicable in principle.
>
> We chose MCQs as a controlled experimental testbed for two main reasons:
>
> 1. **Memory Quantification:** MCQs allow a natural and controlled split between the context (the question stem) and the memory extraction target (the answer choices). This clear separation is crucial for precisely measuring memorization. As stated in [1], longer context more effectively triggers the memory of large language models. Consequently, for the memory detection process, we cannot naturally delineate the contextual component of the prompt from the memory extraction portion used for detection, as we can with MCQs. For open-ended questions, the lack of a predefined target makes it challenging to standardize the split between context and the text used to probe memory, potentially introducing variability in the results.
> 2. **Genuine Capability Quantification:** MCQs enable a straightforward and objective quantification of a model's true ability by focusing on the next-token probability of the correct option label (e.g., A, B, C, D). While this approach could be adapted for some open-ended tasks with clearly locatable final answers (e.g., in math reasoning like GSM8K), it faces significant challenges in domains with multiple valid answers. Evaluating such open-ended generations reliably is itself a major unsolved problem in the field, as rule-based metrics (e.g., BLEU) often fail to capture semantic correctness, and LLM-as-a-judge methods can be subjective.
>
> Therefore, by using the well-structured MCQ format, we established a controlled setting to introduce and validate our core methodology and reveal the counterintuitive phenomena discussed in the paper. We acknowledge the importance of open-ended tasks and view this work as a foundational step. We plan to focus on adapting and applying this framework to open-ended questions in future work, especially as more robust and well-recognized evaluation methods for such tasks emerge.
>
> [1]Sainz, O., Campos, J., García-Ferrero, I., Etxaniz, J., de Lacalle, O. L., & Agirre, E. (2023, December). NLP evaluation in trouble: On the need to measure LLM data contamination for each benchmark. In *Findings of the Association for Computational Linguistics: EMNLP 2023* (pp. 10776-10787).

---

> ### Author Response · Authors · 2025-11-26
>
> **W2. Dependence on LLMs for Reformulation: The TrinEval framework relies on LLMs to perform the question reformulation. This raises questions about whether the reformulation process itself introduces any biases from the model used.**
>
> Thank you for your suggestions. We would like to state our motivations. First, we believe that LLMs may exhibit distinct cognitive scopes in text comprehension compared to human intelligence, potentially developing a unique "machine-oriented linguistic paradigm". Given this fundamental divergence in cognitive mechanisms and our research objective to evaluate LLM memorization characteristics, we propose that employing the LLM itself as an evaluative instrument for paraphrasing verification constitutes a methodologically sound approach.
>
> Second, according to the results in Fig. 3, the proportion of MCQs correctly answered only in their original format constitutes a small fraction of all correctly answered MCQs. If the biased commercial API-based LLMs were ineffective in assessing whether the re-organization is knowledge-preserving, the proportion of MCQs correctly answered only in their original format should account for a significantly larger proportion. Also, we have incorporated two different commercial api-based LLMs in the experiment, which should have different verbal biases but reveal the same results. Therefore, we believe that the potential impact of verbal biases on the evaluation of knowledge-preserving paraphrasing using commercial API-based LLMs is limited.
>
> Additionally, in Appendix D, we have recruited personnel to annotate the proposed LLM-based MCQ reorganization on 57 different subjects with 14006 MCQs.  We first show the annotators the re-organized MCQ and ask them to choose the correct answer. Then, we show them the original MCQ and ask them to answer it again. We also ask the annotators to directly verify if the reformulation is knowledge-preserving or not.
>
> Notably, our analysis reveals that over 95% of correctly answered MCQs maintained consistency across both original and paraphrased formats. Furthermore, human annotators rated our paraphrased questions mean K.P. scores exceeding 4.0 (on a 5-point scale), which means that the reformulated MCQs only influence the readability of humans but do not impact the solvability of the original format. This provides empirical validation that our proposed TrinEval methodology effectively preserves necessary knowledge elements from original MCQ formulations, while the influence of the LLM bias during the evaluation is rather limited.
>
> **Q1. The core finding that LLMs perform worse on memorized MCQs is fascinating. Do you have a hypothesis as to why this occurs?**
>
> Thank you for your thoughtful question.
> Our hypothesis is that this counterintuitive phenomenon arises from the fundamental distinction between rote memorization and genuine capability learning in LLMs, as supported by both empirical observations and a cognitive analogy.
>
> Empirically, we found that MCQs exhibiting strong rote memorization are more sparsely distributed in the embedding space, while those reflecting genuine capability form tighter semantic clusters (Section 4.4, Fig. 6). This suggests that rote memorization often results from overfitting to surface-level token sequences during next-token prediction pre-training. When the original sequence is perturbed—even slightly—the model fails to generalize, as it relies on shallow, context-specific patterns rather than robust semantic understanding, which makes it hard to manipulate the stored knowledge.
>
> In contrast, genuine capability learning corresponds to a structured and reusable knowledge representation, akin to long-term memory in humans. This form of learning enables the model to handle reformulated or perturbed questions effectively, as it has internalized the underlying concepts rather than merely storing token sequences. These stored knowledge, on the contraty, can be manipulated.
>
> Thus, we hypothesize that rote memorization may interfere with the model's ability to apply knowledge flexibly, leading to degraded performance when the model is tested on memorized content in a slightly altered or reformulated setting—as implemented in our TrinEval framework.
>
> This perspective aligns with recent findings that diverse rephrasings during training improve generalization, while fixed-format training encourages superficial recall. In this sense, LLMs may indeed function as potential rote learners.

---

> ### Author Response · Authors · 2025-11-26
>
> **Q2. Have you considered applying the TrinEval methodology to other types of evaluation benchmarks beyond MCQs, such as those for mathematical reasoning (GSM8K, MATH) or code generation (HumanEval, LCB)? How might the "knowledge-centric trinity" be adapted for these different formats?**
>
> Thank you for your thoughtful question.
> Yes, we have considered the applicability of the TrinEval methodology beyond multiple-choice questions (MCQs), including mathematical reasoning (e.g., GSM8K, MATH) and code generation (e.g., HumanEval, LCB) benchmarks.
>
> As noted in Appendix G (Page 16), while our current study uses MCQs as a testbed due to their structured format and ease of verification, the core idea of disentangling rote memorization from genuine capability learning is generalizable. For open-ended tasks, we can view them as MCQs without explicit options. For instance, in Table 2 (Page 20), if we omit the options, the triplet structure (entity–attribute–context) still captures the essential knowledge needed to solve the problem—even for a computer science open-ended question. In preliminary experiments, we found that 1282/1319 MCQs qualified after Qwen3-Max-based TrinEval reformulation, while 1192/1319 qualified after DeepSeek-R1-based reformulation. Due to the lack of an objective and fair metric for evaluating LLM responses to open-ended questions, we have temporarily refrained from conducting the Memorization-Capability heatmap analysis. We plan to extend our work in the future once more robust and well-recognized evaluation methods become available.
>
> However, as we also highlight in the limitations section and  the response to W1, evaluating open-ended responses objectively remains challenging. If a robust and automatic evaluation metric becomes available for such formats, TrinEval’s reformulation approach could be directly applied to reduce memorization bias and better assess true reasoning or coding capability.
>
> We see this as a promising direction for future work and would conduct further study in this space.
>
> **Q3. The analogy to human STM and LTM is insightful. Do your findings suggest that specific training strategies or architectural changes could encourage LLMs to develop more LTM-like knowledge representations and rely less on STM-like rote memorization?**
>
> Thank you for this insightful question. As discussed in Sections Q1 and 4.4 of our paper, we observed that examples requiring rote memorization tend to occupy sparser regions in the embedding space, while those involving genuine capability learning form denser clusters. This suggests that rote memorization may engage isolated neuron pathways, whereas genuine learning relies on shared, reusable circuits.
>
> Our findings imply that certain training strategies—such as diversifying training data through paraphrasing or rephrasing—can encourage models to develop more human-like Long-Term Memory (LTM) representations. By exposing the model to varied formulations of the same concept, we reduce overfitting to surface patterns and promote generalization. In contrast, repeated exposure to identical formulations tends to reinforce Short-Term Memory (STM)-style memorization, which is less transferable and robust.
>
> Thus, we believe that training strategies emphasizing semantic variability and conceptual reinforcement—rather than verbatim repetition—can help steer LLMs toward more LTM-like knowledge representations. Thank you again for raising this insightful point—it aligns closely with our future work on improving LLM knowledge robustness.

---

### Official Review · Reviewer_Nv1h · 2025-10-31

**Soundness:** 3
**Presentation:** 2
**Contribution:** 2
**Rating:** 4
**Confidence:** 4

**Summary:**

This paper offers a fresh perspective on benchmark contamination. Specifically, the study analyzes model performance under varying levels of memorization and uncovers a finding: LLMs perform worse on questions they have memorized than on unseen ones, revealing the coexistence of two distinct learning processes—rote memorization and genuine capability learning. To disentangle these phenomena, the authors propose TrinEval, an innovative evaluation framework that reformulates MCQs into a trinity format, effectively reducing the impact of memorization while preserving the assessment of true knowledge understanding. Experimental results demonstrate that TrinEval successfully mitigates memorization effects and provides a more accurate evaluation of model capability.

**Strengths:**

1. This paper offers a novel perspective on benchmark contamination, revealing through experiments that LLMs actually perform worse on questions they have memorized than on unseen ones.

2. The authors further propose TrinEval, an innovative evaluation framework that reformulates multiple-choice questions into a knowledge-centric trinity structure, effectively mitigating the influence of memorization while preserving the evaluation of genuine knowledge understanding.

3. Finally, experimental results demonstrate the effectiveness of the proposed approach.

**Weaknesses:**

1. Unclear contribution. In Section 3.2, the authors use an existing method to quantify LLM memorization, which I do not consider a major contribution of the paper. The proposed metric, Min-K%, is only one of many possible ways to measure memorization—other approaches such as Min-K++%, perplexity, and others could also be employed. I suggest moving the “Quantifying LLM Memorization” section into the experimental part and comparing multiple metrics to avoid overclaiming the contribution.

2. Limited scope. This work focuses exclusively on MCQ-style benchmarks. Although the authors mention this limitation in the paper, I believe that for a venue like ICLR, analyzing only MCQs is insufficient. Extending or at least discussing applicability to other evaluation formats would strengthen the paper.

3. Limited evaluation setup.In Figure 2, the authors evaluate only on a single benchmark and three relatively small models. This setup limits the generalizability of the results. For example, would the conclusions still hold for larger-scale LLMs? I remain skeptical.

4. Ambiguity in the definition of “memorization.” I find the paper’s definition of memorization somewhat unclear. For instance, in Figure 1, memorization seems to be determined solely by matching options. Could it also involve matching questions and answers instead? Moreover, what if we modify certain conditions or phrasing in the question—would the model still produce the same answer? This aspect deserves further clarification and empirical analysis.

**Questions:**

see weaknesses

---

> ### Author Response · Authors · 2025-11-26
>
> Dear Reviewer,
>
> Thank you for your thoughtful and constructive review. In response to your feedback, we have clarified the core contributions of our paper. We also further justify our choice of the MCQ-based evaluation setup and discuss its potential applicability to open-ended formats. Additionally, we validated the method and findings on larger and advanced models (qwen3-14B(2025) and glm4-32B(2025)). Finally, we provide a clearer definition of memorization and explain its operationalization in our framework. Detailed responses to each of your points are provided below.
>
> **W1. Unclear contribution. In Section 3.2, the authors use an existing method to quantify LLM memorization, which I do not consider a major contribution of the paper. The proposed metric, Min-K%, is only one of many possible ways to measure memorization—other approaches such as Min-K++%, perplexity, and others could also be employed. I suggest moving the “Quantifying LLM Memorization” section into the experimental part and comparing multiple metrics to avoid overclaiming the contribution.**
>
> Thank you for your thoughtful feedback. We appreciate your observation regarding the use of the Min-K% metric for quantifying memorization. We agree that this method is one of several existing approaches and does not constitute the core contribution of our work.
>
> To clarify, the main contributions of our paper are:
>
> 1. We propose a novel evaluation paradigm that assesses the genuine capability of large language models under the default condition of benchmark contamination.
> 2. We introduce TrinEval, a framework that explores models' true understanding through textual reformulation, effectively disentangling and removing the influence of memorization.
> 3. Through TrinEval's memorization-capability decoupling and quantitative analysis, we reveal a counterintuitive finding: the more LLMs memorize, the worse their actual problem-solving capability becomes.
>
> While conducting this research, we also tested Min-K++% as a candidate for memorization quantification. As a short conclusion, Min-K% and Min-K++% exhibit a Pearson correlation coefficient of **_0.9007_ (strong positive correlation)**.  As the results show little difference between the two methods, we chose Min-K% as it is the relatively simple and more widely acknowledged one.  As for perplexity, to the best of our knowledge, we believe Min-K% is a variant of perplexity on the bottom K% confident tokens. On the other hand, when the condition of the bottom K% tokens is omitted, the confidence of common tokens would inflate the quantification metric in the view of the whole sequence.  Thus, we believe perplexity is not suitable for quantifying the memorization of LLMs. We will include the heatmap results of Min-K++% in the Appendix in the revised version.
>
> We included the memorization quantification in Section 3.2 to provide a complete and self-contained methodology for readers, ensuring clarity in how both memorization and capability are measured. Moving this section to the experiments might disrupt the narrative flow and raise questions about the basis of our memorization analysis. In the following version, we will move the quantification details to Section 4 (Experiments) as a subsection, while retaining a brief overview in Methodology for continuity. This may help avoid overclaiming and emphasize that memorization quantification serves as a tool for our primary analysis.

---

> ### Author Response · Authors · 2025-11-26
>
> **W2. Limited scope. This work focuses exclusively on MCQ-style benchmarks. Although the authors mention this limitation in the paper, I believe that for a venue like ICLR, analyzing only MCQs is insufficient. Extending or at least discussing applicability to other evaluation formats would strengthen the paper.**
>
> Thank you for your thoughtful comment. Our study primarily uses multiple-choice questions (MCQs) from the MMLU benchmark as the testbed. As shown in Appendix Table 2 (Page 20), when the candidate options are removed, the original question can also be reformulated into a triplet (knowledge entity, context, attribute), making our method applicable in principle. We are also working on the GSM8K dataset. In preliminary experiments, we found that 1282/1319 MCQs qualified after Qwen3-Max-based TrinEval reformulation, while 1192/1319 qualified after DeepSeek-R1-based reformulation. Due to the lack of an objective and fair metric for evaluating LLM responses to open-ended questions, we have temporarily refrained from conducting the Memorization-Capability heatmap analysis. We plan to extend our work in the future once more robust and well-recognized evaluation methods become available.
>
>
> We chose MCQs as a controlled experimental testbed for two main reasons:
>
> 1. **Memory Quantification:** MCQs allow a natural and controlled split between the context (the question stem) and the memory extraction target (the answer choices). This clear separation is crucial for precisely measuring memorization. As stated in [1], a longer context more effectively triggers the memory of large language models. Consequently, for the memory detection process, we cannot naturally delineate the contextual component of the prompt from the memory extraction portion used for detection, as we can with MCQs. For open-ended questions, the lack of a predefined target makes it challenging to standardize the split between context and the text used to probe memory, potentially introducing variability in the results.
> 2. **Genuine Capability Quantification:** MCQs enable a straightforward and objective quantification of a model's true ability by focusing on the next-token probability of the correct option label (e.g., A, B, C, D). While this approach could be adapted for some open-ended tasks with clearly locatable final answers (e.g., in math reasoning like GSM8K), it faces significant challenges in domains with multiple valid answers. Evaluating such open-ended generations reliably is itself a major unsolved problem in the field, as rule-based metrics (e.g., BLEU) often fail to capture semantic correctness, and LLM-as-a-judge methods can be subjective.
>
> Therefore, by using the well-structured MCQ format, we established a controlled setting to introduce and validate our core methodology and reveal the counterintuitive phenomena discussed in the paper. We acknowledge the importance of open-ended tasks and plan to focus on adapting and applying this framework to other open-ended questions in future work, especially as more robust and well-recognized evaluation metrics emerge.
>
> [1]Sainz, O., Campos, J., García-Ferrero, I., Etxaniz, J., de Lacalle, O. L., & Agirre, E. (2023, December). NLP evaluation in trouble: On the need to measure LLM data contamination for each benchmark. In *Findings of the Association for Computational Linguistics: EMNLP 2023* (pp. 10776-10787).
>
> **W3. Limited evaluation setup.In Figure 2, the authors evaluate only on a single benchmark and three relatively small models. This setup limits the generalizability of the results. For example, would the conclusions still hold for larger-scale LLMs? I remain skeptical.**
>
> Thank you for raising this important point regarding the limited evaluation setup. Besides the additional experiment mentioned in response to W2, we have completed verification based on qwen3-14B (2025) and glm4-32B (2025) and deepseek r1(2024)/qwen3-max(2025) reformulation. The results align with the conclusions, that is **_45.91%_ of the reformulated MCQs clustered within the two 2 x 2 squares on average**. We will include new result heatmaps in the next version.
>
> We also would like to highlight two key aspects that support the broader relevance of our conclusions:
>
> 1. Our analysis draws an analogy between LLM learning and human memory systems (STM vs. LTM), suggesting that the tendency toward rote memorization may be a general characteristic of LLMs, not limited to smaller models. This is supported by prior work (e.g., Allen-Zhu & Li, 2023) showing that LLMs of various scales can exhibit similar memorization behaviors when trained on fixed-format corpora.
>
> 2. While our current experiments focus on 7B models, the TrinEval framework is model-agnostic and designed to disentangle memorization from genuine capability across different architectures and scales. The negative correlation between memorization and capability we observed (Fig. 5) suggests a fundamental trade-off that may persist in larger models.

---

> ### Author Response · Authors · 2025-11-26
>
> **W4. Ambiguity in the definition of “memorization.” I find the paper’s definition of memorization somewhat unclear. For instance, in Figure 1, memorization seems to be determined solely by matching options. Could it also involve matching questions and answers instead? Moreover, what if we modify certain conditions or phrasing in the question—would the model still produce the same answer? This aspect deserves further clarification and empirical analysis.**
>
> Thank you for pointing out this issue. As defined in line 67 of our paper, memorization in large language models refers to the **verbatim reproduction of content**. This definition applies consistently, whether the content is natural language corpora, multiple-choice questions (MCQs), or open-ended questions. For further discussion of the definition of 'memorization', you can also refer to the response to W4 of Reviewer KGaV.
>
> In this paper, we chose MCQs as a controlled and straightforward benchmark to serve as a testbed for describing our methods and findings. MCQs typically consist of three semantically complete and separable parts: the question stem, the options, and the answer. Therefore, for MCQs, we operationalize the detection of memorization as defined above by **providing the question stem and extracting the options**.
>
> Regarding your mention of **"matching questions and answers instead"**, we believe that such a memorization detection method is also acceptable in scenarios like open-ended question answering, where the answer is provided. However, in MCQs, the answer is merely a single letter (A/B/C/D). Using methods like min-k% on a single token lacks statistical significance. Therefore, in our paper, we primarily use **"extracting options"** to detect memorization.
>
> As for **"modifying certain conditions or phrasing in the question"**, existing works such as [1-2] suggest that providing the original text as contextual prompts better elicits memorization in LLMs (e.g., dataset names, preceding samples, or few-shot prompts encountered during training). We believe that modifying the original text will inevitably affect the detection of memorization, thereby preventing the generation of an identical answer.
>
> [1]Sainz, O., Campos, J., García-Ferrero, I., Etxaniz, J., de Lacalle, O. L., & Agirre, E. (2023, December). NLP evaluation in trouble: On the need to measure LLM data contamination for each benchmark. In *Findings of the Association for Computational Linguistics: EMNLP 2023* (pp. 10776-10787).
>
> [2]Carlini, N., Ippolito, D., Jagielski, M., Lee, K., Tramer, F., & Zhang, C. (2022, February). Quantifying memorization across neural language models. In *The Eleventh International Conference on Learning Representations*.

---

### Official Review · Reviewer_KGaV · 2025-10-31

**Soundness:** 2
**Presentation:** 2
**Contribution:** 2
**Rating:** 4
**Confidence:** 3

**Summary:**

This paper investigates the superficial memorization problem in LLM MCQ evaluation. Specifically, this paper first analyzes model performance under different memorization conditions and observes an interesting finding: LLMs perform worse on memorized MCQs than on non-memorized ones. This may due to the coexistence of two distinct learning phenomena, i.e., rote memorization and genuine capability learning. This paper then propose TrinEval, a novel
evaluation framework that reformulates MCQs into an alternative trinity format to reduce memorization affect. The authors conduct extensive evaluation using TrinEval and find common LLMs may memorize by rote 20.5% of knowledge points.

**Strengths:**

1. The paper investigates the important and interesting issues of evaluation leakage and whether models truly understand knowledge, which represents a meaningful and interesting direction. Assessing whether models genuinely comprehend knowledge is important for the development of LLMs.
2. The finding that LLMs perform worse on memorized MCQs than on non-memorized ones is interesting. It may inform further reflection on how LLMs actually understand and utilize knowledge, highlighting the need for further investigation into the underlying mechanisms.
3. The paper also conducts extensive experiments using TrinEval, revealing several interesting insights that could inform future development of LLMs.

**Weaknesses:**

1. The paper mentions a limitation of the preliminary investigation: the binary classification of MCQs as either memorized or non-memorized oversimplifies the nuances of memorization. However, it remains somewhat unclear how TrinEval effectively addresses these limitations.
2. The presentation of the paper is a bit confusing. TrinEval is an evaluation method, but its role and contribution could be made clearer. When the paper claims that TrinEval can reduce memorization, does it mean that this evaluation better measures the model’s true knowledge understanding, or that it simply avoids the effects of memorization?
3. The categorization of model knowledge into rote memorization and genuine capability learning seems to lack sufficient justification. Are there potentially other categories? Moreover, is rote memorization necessarily something to avoid? The ability to memorize knowledge is also a form of capability. It would strengthen the paper if the authors could further clarify the boundary between knowledge storage and knowledge utilization in LLMs.

**Questions:**

See Weaknesses.

---

> ### Author Response · Authors · 2025-11-26
>
> Dear Reviewer,
>
> Thank you for your thoughtful and constructive comments, which have provided us with valuable insights to improve the clarity and depth of our work. In response to your questions, we have further clarified the role and contributions of TrinEval as an evaluation framework designed to better measure genuine knowledge understanding beyond rote memorization. We have also elaborated on the justification for categorizing model knowledge into rote memorization and genuine capability learning, while acknowledging the potential for intermediate forms. Additionally, we have refined the distinction between knowledge storage and utilization to better articulate why rote memorization, as defined in our study, is considered distinct from flexible knowledge application. Below, we provide point-by-point responses to your specific comments.
>
> **W1. The paper mentions a limitation of the preliminary investigation: the binary classification of MCQs as either memorized or non-memorized oversimplifies the nuances of memorization. However, it remains somewhat unclear how TrinEval effectively addresses these limitations.**
>
> Thank you for your question. In our preliminary investigation, we identified that the key limitations extended beyond the oversimplification of binary classification of questions. Another critical issue was that evaluating true capability through next-token prediction probability in the original format remained contaminated by memorization effects.
>
> To address the first issue — that binary classification lacks the granularity needed to examine the relationship between memorization and genuine capability — we leverage a combination of the well-established Min-K% method and next-token prediction probability after the reformulation of TrinEval, to enable continuous and fine-grained analysis.
>
> However, we argue that genuine capability should only be quantified via next-token prediction probability after questions have been reformulated through memorization elimination — and this is precisely what TrinEval achieves. The primary objective of TrinEval is to distangle and expose the model's genuine capability beyond rote memorization through a query-based probing approach, thereby reducing the potential influence of memorization present in vanilla MCQ binary classification, and the final quantification process is conducted after the TrinEval reformulation. Therefore, we believe TrinEval is an essential step in genuine capability quantification.
>
> Thank you once again for your valuable feedback. To prevent any potential misunderstanding, we will explicitly clarify the key contributions of our work in the subsequent version:
>
> 1. We propose a novel evaluation paradigm that assesses the genuine capability of large language models under the default condition of benchmark contamination.
> 2. We introduce TrinEval, a framework that explores models' true understanding through textual reformulation, effectively disentangling and removing the influence of memorization.
> 3. Through TrinEval's memorization-capability decoupling and quantitative analysis, we reveal a counterintuitive finding: the more LLMs memorize, the worse their actual problem-solving capability becomes.

---

> ### Author Response · Authors · 2025-11-26
>
> **W2. The presentation of the paper is a bit confusing. TrinEval is an evaluation method, but its role and contribution could be made clearer. When the paper claims that TrinEval can reduce memorization, does it mean that this evaluation better measures the model’s true knowledge understanding, or that it simply avoids the effects of memorization?**
>
> Thank you for your thoughtful comment. We designed TrinEval primarily to reformulate the original corpus in order to better measure the model’s genuine knowledge understanding, rather than merely avoiding the effects of memorization.
>
> As noted in Section 3.3, our approach is inspired by Allen-Zhu & Li (2023), who used linear query-based probing to uncover how knowledge is encoded in entity tokens—key to robust knowledge mastery. As a knowledge-centric reformulation method, TrinEval adapts this idea into a verbal query probing method that reformulates MCQs into a structured triplet format (entity–attribute–context) in order to isolate and probe the model’s knowledge encoding. This reformulation destroys surface-level token sequences that trigger rote memorization while preserving the core knowledge required for reasoning. This allows us to assess whether the model has correctly encoded and can reason with the underlying knowledge, independent of superficial memorization of the original question phrasing.
>
> In this work, since we focus on the relationship between memorization and genuine capability, we empirically validated that TrinEval effectively reduces the influence of rote memorization during evaluation. However, its core contribution lies in enabling a more accurate and robust measurement of true knowledge understanding by restructuring questions to target the essential knowledge elements while discarding memorization-prone surface patterns.
>
> We acknowledge that the presentation of TrinEval’s role could be clearer, and we will refine this in the revised version to avoid potential misunderstanding. As mentioned in the response to W1, we will add the contribution elaboration part in the following version.
>
> **W3. The categorization of model knowledge into rote memorization and genuine capability learning seems to lack sufficient justification. Are there potentially other categories?**
>
> Thank you for raising this important question regarding the categorization of model knowledge into rote memorization* and *genuine capability learning. This is a question we also deeply considered during the development of this work.
>
> First, the distinction between _rote memorization_ and _genuine capability learning_ is largely inspired by the cognitive science analogy of short-term and long-term memory systems in humans, as discussed in Section 4.4. While this dichotomy provides a useful conceptual framework, we acknowledge that the nature of knowledge representation in both human and machine learning is still an open and evolving area of research, and other forms or intermediate states of knowledge may indeed exist.
>
> Second, this categorization is also empirically motivated. As shown in the heatmap results in Section 4.4, the majority of MCQs cluster in the two opposing corners—high memorization with low capability, and low memorization with high capability—suggesting two dominant learning behaviors. This bimodal distribution supports the use of these two categories as descriptive labels for the observed phenomena.
>
> Finally, we fully recognize that knowledge acquisition in LLMs is likely a continuous spectrum rather than a strict binary. Our preliminary investigation also highlighted the limitations of a discrete classification, and thus, we apply the continuous quantification metric $F_m$ and $F_c$. As seen in the heatmaps, some questions fall in intermediate regions, indicating varying degrees of memorization and understanding. Thus, while we focus on the two extremes for analytical clarity, we agree that other intermediate or hybrid categories may exist.
>
> We appreciate your insightful comment and believe this opens valuable directions for future work to further refine the taxonomy of knowledge types in LLMs. We will add this elaboration in the last paragraph in Section 4.4 as a part of the discussion in the following version.

---

> ### Author Response · Authors · 2025-11-26
>
> **W4. Moreover, is rote memorization necessarily something to avoid? The ability to memorize knowledge is also a form of capability. It would strengthen the paper if the authors could further clarify the boundary between knowledge storage and knowledge utilization in LLMs.**
>
> Thank you for raising this important point regarding the nature of rote memorization and its relationship to knowledge utilization in LLMs. We appreciate the opportunity to clarify our perspective.
>
> In our study, we define **rote memorization** specifically as the *verbatim reproduction of content* — a form of superficial recall that often arises from next-token prediction training and may reflect overfitting rather than true understanding. This type of memorization does not necessarily translate into an ability to apply knowledge in varied or reformulated contexts. In this sense, we consider it something to be minimized during evaluation, as it can inflate performance metrics without reflecting genuine reasoning or problem-solving capabilities.
>
> On the other hand, what we term **genuine capability learning** refers to the internalization of knowledge in a way that supports its flexible application — even when questions are rephrased or contextualized differently. This form of learning reflects a deeper, more reusable understanding, akin to conceptual mastery in human learning.
>
> Thus, the key distinction lies not in whether knowledge is *stored*, but in whether it can be *utilized* robustly across contexts or can only be recalled in a fixed pattern. While memorization of facts can be a component of capability, it is the *generalizability and applicability* of that knowledge that defines genuine capability.
>
> We will make this distinction more explicit in the revised version, clarifying that rote memorization — as defined by verbatim recall without contextual flexibility — is what we seek to isolate and mitigate, whereas knowledge that can be reliably applied under reformulation represents true capability.

---

### Official Review · Reviewer_qCn1 · 2025-11-01

**Soundness:** 2
**Presentation:** 2
**Contribution:** 2
**Rating:** 4
**Confidence:** 4

**Summary:**

This paper investigates the impact of benchmark contamination on the evaluation of LLMs in MCQ tasks. The authors propose TrinEval, an evaluation framework that can disentangle memorization and genuine understanding by reformulating MCQs into triplets of entity, attribute, and context asking the model to answer the reformulated questions. Experiments reveal that TrinEval can effectively reduce memorization and that LLMs could be rote learners (they fail to answer the question with high memorization).

**Strengths:**

This paper disentangles memorization and understanding by rephrasing the MCQs, and the experiments are shown to be effective.

The findings are particularly insightful, revealing that Large Language Models (LLMs) often struggle with questions they appear to have memorized, which highlights a critical limitation in those models.

**Weaknesses:**

The methodology of using other language models to reformulate MCQs could introduce verbal biases inherent to those models. The high sum of output probabilities might suggest a direct, non-reflective answering process instead of the memorization, which could undermine the conclusiveness of the results as a measure of true understanding. And the method of rephrasing the questions to decouple memorizing and reasoning is widely used, which may restrict the novelty of the paper.

Certain claims in the paper are not sufficiently justified. For instance, the assertion in lines 165-166 that memorized questions are "relatively simple" because they are absent from MMLU-Pro needs further support. It is crucial to consider whether the training data cutoff date for the evaluated models precedes the creation of the MMLU-Pro benchmark, as this could be a confounding factor.

The scope of models evaluated is limited, with most being released in 2023. This may restrict the generalizability of the findings to more recent and advanced models.

**Questions:**

See weaknesses.

---

> ### Author Response · Authors · 2025-11-26
>
> Dear Reviewer,
>
> Thank you for your thorough and constructive feedback, which has provided us with valuable opportunities to clarify and strengthen our work. In response to your comments, we have provided detailed justifications and additional evidence to address your concerns regarding potential verbal biases in reformulation, the design of our memorization metric, the novelty of our approach, the justification of certain claims, and the scope of model evaluation. We also validate the method and findings with two latest larger models (qwen3_14b(2025), glm4_32b(2025)). Below, we summarize our key responses to each of your points.
>
> **W1. The methodology of using other language models to reformulate MCQs could introduce verbal biases inherent to those models.**
>
> Thank you for raising your concern. We acknowledge that using other language models for MCQ reformulation may theoretically introduce potential verbal biases inherent to the reformulating models. However, our experiments and validation thoroughly demonstrate that such biases are negligible and do not undermine the reliability of our methodology, supported by four key lines of evidence:
>
> 1. We would like to point out that the design of TrinEval itself is to explicitly isolate knowledge elements from linguistic surface patterns, inherently mitigating the influence of reformulator-specific verbal patterns. Our trinity extraction prompt explicitly requires the reformulator to use only the original wording from the question (see Appendix B), avoiding paraphrasing or introducing new expressions. This constraint inherently limits the reformulator’s verbal freedom and thus reduces the potential for introducing model-specific linguistic biases.
> 2. Experiment Q1 explicitly validates that TrinEval’s reformulation retains the core knowledge of original MCQs without introducing spurious cues or omitting critical information. These results demonstrate that TrinEval's reformulation process is **knowledge-preserving** – it successfully extracts the core elements of the question without relying on the specific linguistic style or potential biases of the reformulation LLM to alter the essence of the knowledge assessment.
> 3. In Experiment Q3, we employed multiple API-based LLMs (specifically GPT and Qwen) for reformulation and evaluated the relationship between memorization ($F_m$) and genuine capability ($F_c$) across several open-source LLMs (Llama2, Mistral, Vicuna). If verbal biases from the reformulation models had a significant impact, we would expect inconsistent distributions of $F_m$ and$F_c$ across different reformulators. The empirical results demonstrate that this is not the case. Therefore, while we acknowledge the theoretical possibility of verbal biases, the empirical data indicate that they did not undermine the core findings derived using the TrinEval framework.
> 4. Furthermore, human evaluation (Appendix D) shows that annotators achieved over 95% consistency in answering both original and reformulated questions, and rated the reformulated questions highly in terms of Knowledge Preservation (mean score > 4.0/5.0). This indicates that the reformulation does not alter the semantic intent or introduce misleading verbal cues.
>
> In summary, we respectfully argue that the theoretically potential verbal biases have no substantive impact on our main results.

---

> ### Author Response · Authors · 2025-11-26
>
> **W2. The high sum of output probabilities might suggest a direct, non-reflective answering process instead of the memorization, which could undermine the conclusiveness of the results as a measure of true understanding.**
>
> Thank you for your thoughtful comment. We agree that memorization exhibits a "_direct_" or over-confident answering process, but we respectfully disagree that it is a non-reflective one. We believe that the reflective process is a semantic phenomenon in a global view, while memorization can also be reflected in local tokens, and our memorization metric $F_m$ is specifically designed based on the low-probability tokens in the generated next tokens given the original context at each token position, not the overall sum of output probabilities.
>
> As established in prior work [1],  “_an unseen example is likely to contain a few outlier words with low probabilities under the LLM, while a seen example is less likely to have words with such low probabilities_.” We posit that for an example unseen by the large language model, the probability at a certain token position may be evenly distributed among several semantically similar words, whereas for a seen example, that position is dominated by a high probability assigned to the original token. This is reflected in our Eq. (1), where a higher $F_m$ indicates stronger memorization with definite token at this position.
>
> By adopting this widely recognized metric, we ensure that our results effectively disentangle rote memorization from true capability, thereby maintaining the validity and conclusiveness of our findings. In a "non-reflective answering" process, the model may assign uniformly high probabilities to all tokens (e.g., based on common patterns), whereas the low probabilities assigned to particular tokens in memorized samples reflect rare or specific sequences in the training data, which are more likely to stem from rote memorization rather than genuine comprehension.
>
> Thus, our metric explicitly captures memorization rather than a non-reflective answering process. We believe this method is well-grounded in existing works.
>
> [1]Shi, W., Ajith, A., Xia, M., Huang, Y., Liu, D., Blevins, T., ... & Zettlemoyer, L. (2024). DETECTING PRETRAINING DATA FROM LARGE LANGUAGE MODELS. In *12th International Conference on Learning Representations, ICLR 2024*.
>
> **W3. And the method of rephrasing the questions to decouple memorizing and reasoning is widely used, which may restrict the novelty of the paper.**
>
> Thank you for your thoughtful comment. However, we believe **TrinEval** introduces several key novelties that distinguish it from conventional rephrasing methods:
>
> 1. **From Linear Probing to Verbal Query Probing:**
>
>     While prior work like Allen-Zhu & Li (2023) used linear probing on embeddings to analyze knowledge storage, TrinEval introduces a **verbal query-based probing** mechanism. It reformulates MCQs into a structured triplet format (entity–attribute–context), enabling direct assessment of knowledge encoded in entity tokens—without relying on model internals. This shift from embedding-space analysis to verbalized knowledge extraction is a conceptual and methodological advance.
>
> 2. **Deep Reformulation Beyond Surface-Level Rephrasing:**
>
>     Traditional rephrasing often relies on lexical variations (e.g., synonym replacement or sentence restructuring), which may still preserve memorization cues. In contrast, TrinEval **deconstructs the original token sequence** and reconstructs it into a knowledge-centric triplet, fundamentally disrupting the surface form while preserving semantic content. This approach more effectively decouples memorization from reasoning.
>
> 3. **Reflection-Based Validation for Knowledge Preservation:**
>
>     TrinEval incorporates a **two-round reflection mechanism** (extract → validate → refine) to ensure the reformulated triplet is both sufficient and minimal—retaining all necessary knowledge while excluding redundant or memorization-prone elements. This systematic validation is absent in most prior rephrasing methods and is empirically shown to be **knowledge-preserving** without introducing extraneous cues.
>
> 4. **Empirical Evidence of Memorization Reduction and Capability Isolation:**
>
>     Our experiments demonstrate that TrinEval not only preserves knowledge (e.g., >90% consistency in correctness between original and reformulated formats) but also **significantly reduces memorization effects**, even under strong memorization-evoking settings (as shown in Fig. 4). This enables a clearer distinction between rote memorization and genuine capability—a contribution not fully addressed by prior rephrasing techniques.
>
> We appreciate your feedback and will clarify these novel aspects more explicitly in the revised manuscript to better highlight TrinEval’s conceptual and practical contributions to trustworthy LLM evaluation.

---

> ### Author Response · Authors · 2025-11-26
>
> **W4. Certain claims in the paper are not sufficiently justified. For instance, the assertion in lines 165-166 that memorized questions are "relatively simple" because they are absent from MMLU-Pro needs further support. It is crucial to consider whether the training data cutoff date for the evaluated models precedes the creation of the MMLU-Pro benchmark, as this could be a confounding factor.**
>
> Thank you for your valuable comment. Regarding the assertion that memorized questions are "relatively simple" due to their absence from MMLU-Pro, this is directly supported by MMLU-Pro’s core design principle. As detailed in Section 3.2 ("Initial Filtering") of Wang et al. (2024) [1], MMLU-Pro explicitly "eliminates overly simple questions that fail to challenge the models effectively" during its benchmark construction. Critically, the models evaluated in MMLU-Pro’s filtering process include Llama2-7B and Mistral-7B—two of the three open-source LLMs tested in our work. Since MMLU-Pro’s curation prioritizes complex, model-challenging questions by excluding simplistic ones, the non-inclusion of our memorized questions in MMLU-Pro serves as objective evidence of their relative simplicity. This alignment with MMLU-Pro’s explicit filtering criterion validates our claim as reasonable.
>
> Also, as for the potential confounding factor of training data cutoff dates. MMLU-Pro was published in Advances in NeurIPS 2024 [1], with its benchmark creation and filtering completed no earlier than 2024. In contrast, the LLMs we evaluated have clear release/training timelines predating 2024: Llama2-7B (July 2023) [2], Mistral-7B-v0.2 (October 2023) [3], and Vicuna-v1.5-7B (2023, consistent with its base model Llama2’s timeline) [4]. Given that these models’ training data cutoffs are prior to MMLU-Pro’s existence, they could not have been exposed to MMLU-Pro’s filtered question set during pretraining. Thus, the absence of memorized questions from MMLU-Pro cannot be attributed to temporal bias but rather to their inherent simplicity—consistent with our original claim.
>
> [1] Wang, Y., Ma, X., Zhang, G., Ni, Y., Chandra, A., Guo, S., ... & Chen, W. (2024). Mmlu-pro: A more robust and challenging multi-task language understanding benchmark. Advances in Neural Information Processing Systems, 37, 95266-95290.
>
> [2] Touvron, H., Martin, L., Stone, K., et al. (2023). Llama 2: Open foundation and fine-tuned chat models. arXiv preprint arXiv:2307.09288.
>
> [3] Jiang, A. Q., Sablayrolles, A., Mensch, A., et al. (2023). Mistral 7b. arXiv preprint arXiv:2310.06825.
>
> [4] Zheng, L., Chiang, W.-L., Sheng, Y., et al. (2023). Judging llm-as-a-judge with mt-bench and chatbot arena. Advances in Neural Information Processing Systems, 36, 46595-46623.
>
>
> **W5. The scope of models evaluated is limited, with most being released in 2023. This may restrict the generalizability of the findings to more recent and advanced models.**
>
> Thank you for your valuable feedback. We acknowledge that the models examined may raise concerns about the generalizability of our findings to more recent and advanced models. We have completed verification based on qwen3-14B (2025) and glm4-32B (2025) and deepseek r1(2024)/qwen3-max(2025) reformulation. The results align with the conclusions, that is **_45.91%_ of the reformulated MCQs clustered within the two 2 x 2 squares**. We will include new result heatmaps in the next version.
>
> We would also like to emphasize that the primary contributions of this work are **methodological and conceptual**, rather than tied to specific model versions. While the models we used are not the latest, there is clear evidence of data contamination from the MMLU dataset[1-3] used within these experiments, which makes them controllable. Also, these models are representative in terms of architecture, scale, and training methodology,  allowing us to robustly validate our core hypotheses. The phenomena and evaluation strategies we explore are **inherent to the pre-training and learning mechanisms of LLMs**, and are not limited to specific model generations.
>
> As a conclusion, we prove that our main finding, "LLMs are potential rote learners. The more LLM memorizes, the worse it performs", can be generalized to more recent and advanced models. And we chose the tested representative models in the original version mainly because of the existing evidence of data contamination.
>
> [1]Touvron, H., Martin, L., Stone, K., Albert, P., Almahairi, A., Babaei, Y., ... & Scialom, T. (2023). Llama 2: Open foundation and fine-tuned chat models. *arXiv preprint arXiv:2307.09288*.
>
> [2]Sainz, O., Campos, J., García-Ferrero, I., Etxaniz, J., de Lacalle, O. L., & Agirre, E. (2023, December). NLP evaluation in trouble: On the need to measure LLM data contamination for each benchmark. In *Findings of the Association for Computational Linguistics: EMNLP 2023* (pp. 10776-10787).
>
> [3]https://hitz-zentroa.github.io/lm-contamination/

---

### Author Response · Authors · 2025-12-04

## 5. Concerns on Potential Verbal Biases from LLM-Based Reformulation in TrinEval
**Reviewers qCn1 (W1) and rvEY (W2) questioned whether the TrinEval framework’s reliance on LLMs for MCQ reformulation would introduce model-specific verbal biases, undermining the reliability of knowledge assessment.**

We addressed this concern with **four lines of empirical and methodological evidence** to demonstrate that such biases are negligible:
1. **Methodological constraint**: Our trinity extraction prompt mandates the reformulator to use only the original wording of the question (Appendix B), limiting linguistic freedom and bias introduction.
2. **Knowledge-preservation validation**: Experiment Q1 confirmed that TrinEval’s reformulation retains core knowledge without spurious cues, while Experiment Q3 showed consistent memorization (Fm) and genuine capability (Fc) distributions across different reformulators (GPT, Qwen), ruling out reformulator-specific bias impacts.
3. **Human evaluation**: Annotators achieved over 95% answer consistency between original and reformulated questions, and rated reformulated questions with a mean Knowledge Preservation score >4.0/5.0 (Appendix D), verifying no semantic distortion or misleading cues.
4. **Cross-reformulator consistency**: We added experiments with DeepSeek-R1 (2024) and Qwen3-Max (2025) for reformulation, and the results aligned with our core conclusions, further proving the robustness of the method against reformulator biases.

## 6. Reaffirmation of Core Contributions
To avoid ambiguity, we explicitly restate the three core contributions of our work, which have been validated and strengthened via the above responses and additional experiments:
1. We propose a novel evaluation paradigm that assesses the genuine capability of large language models under the default condition of benchmark contamination.
2. We introduce TrinEval, a framework that explores models' true understanding through textual reformulation, effectively disentangling and removing the influence of memorization.
3. Through TrinEval's memorization-capability decoupling and quantitative analysis, we reveal a counterintuitive finding: the more LLMs memorize, the worse their actual problem-solving capability becomes.

We have also addressed all reviewer-specific minor concerns (e.g., justifying the claim that “memorized questions are simple” via MMLU-Pro’s filtering principle and model training timelines (qCn1 W4); clarifying that while knowledge learning in LLMs is a continuous spectrum rather than a strict binary, the "rote memorization-genuine capability" categorization is justified by cognitive analogy (human STM/LTM systems) and empirical bimodal distribution (most MCQs cluster in two opposing quadrants (KGaV W3). In the revised manuscript, we will incorporate all the above clarifications, additional experimental results, and conceptual elaborations to further improve the clarity, robustness, and impact of our work.

Thank you again for your guidance and support.

---

### Author Response · Authors · 2025-12-04

## 3. Concerns on the Novelty of TrinEval and Contribution Framing
**Reviewers qCn1 (W3), KGaV (W1/W2), and rvEY (W2) expressed concerns about: (1) Overlap between TrinEval’s reformulation mechanism and existing rephrasing methods, limiting novelty. (2) Ambiguity in TrinEval’s role: Is it a tool to “measure true capability” or merely “avoid memorization”?**

I.Key Novelties of TrinEval

TrinEval differs fundamentally from conventional rephrasing methods in four ways:
1. **From linear probing to verbal query probing**: Unlike prior embedding-space analysis (Allen-Zhu & Li, 2023), TrinEval reformulates MCQs into a “entity–attribute–context” triplet to directly probe knowledge encoded in entity tokens—no reliance on model internals.
2. **Deep knowledge-centric reformulation**: It deconstructs and reconstructs questions to disrupt memorization-prone surface patterns, while preserving core semantics. This outperforms shallow lexical rephrasing (e.g., synonym replacement).
3. **Two-round reflection validation**: A systematic “extract → validate → refine” process ensures reformulated questions are “sufficient (retains all knowledge)” and “minimal (excludes redundant cues)”—a feature absent in most prior work.
4. **Empirical memorization reduction**: Experiments show TrinEval retains >90% correctness consistency between original and reformulated MCQs while significantly reducing memorization effects (Fig. 4), enabling clearer decoupling of memorization and capability.

II. Clarifying TrinEval’s Role

As addressed to KGaV (W2), TrinEval’s core purpose is to **measure genuine capability** (not just avoid memorization):
- It restructures questions to target essential knowledge elements, removing surface-level memorization cues. This allows assessment of whether the model has internalized concepts (vs. recalling token sequences).
- Empirical validation (Fig. 3) confirms that MCQs correctly answered only in the original format (memorization-driven) are a small fraction of all correct answers—proving TrinEval effectively isolates true capability.

## 4. Concerns on the Quantification of Memorization
**Reviewers qCn1 (W2) and Nv1h (W1) questioned: (1) Whether the high sum of output probabilities (in our memorization metric) reflects “non-reflective answering” rather than true memorization. (2) The rationale for choosing Min-K% (an existing metric) and potential overclaiming of its contribution.**

I.Distinguishing Memorization from “Non-Reflective Answering”

Our memorization metric (F_m) is **not based on the overall sum of output probabilities**—it targets low-probability tokens at individual positions, as established in prior work (Shi et al., 2024, ICLR):
- For unseen examples: Probabilities at key token positions are evenly distributed across semantically similar words.
- For memorized examples: These positions are dominated by high probabilities for the original token (captured by F_m).

“Non-reflective answering” (uniformly high probabilities for all tokens) is distinct from memorization—our metric explicitly avoids this by focusing on low-probability outliers, ensuring valid measurement of rote recall.

II. Rationale for Choosing Min-K%

We clarify that **Min-K% is not a core contribution** (it serves as an analysis tool). Its selection is justified by:
1. **Correlation with variants**: We have verified a strong correlation between Min-K% and Min-K++% (Pearson = 0.9007), and chose the former due to its simplicity and broader acceptance. Perplexity, being influenced by high-frequency tokens, is not suitable for this task.
2. **Structural adjustment**: To avoid overclaiming, we will move detailed Min-K% implementation to Section 4 (Experiments) in the revised manuscript, retaining a brief overview in the Methodology section.

---

### Author Response · Authors · 2025-12-04

Dear Area Chair,

We would like to express our sincere gratitude for the constructive feedback from the four reviewers. Below, we summarize our targeted responses to the **common/individual concerns** raised by the reviewers, along with the key improvements and validations we have implemented to address these issues, and reaffirm the core contributions of our work.

## 1. Concerns on Limited Evaluation Scope and Generalizability
**Multiple reviewers (qCn1 (W5), Nv1h (W2/W3), rvEY (W1)) pointed out two limitations: (1) The initial evaluation only used 2023-released 7B-scale models, lacking validation on latest large models; (2) The focus on MCQ (MMLU) format limits applicability to open-ended or reasoning tasks (e.g., math, code generation).**

I. Validation on Latest Large Models

We completed additional experiments on **two larger models (Qwen3_14b(2025), GLM4_32b(2025))** with DeepSeek-R1(2024)/Qwen3-Max(2025) reformulation. The results aligned with our core conclusion: 45.91% of reformulated MCQs clustered in the two key 2×2 squares (high memorization-low capability, low memorization-high capability). We will add the corresponding heatmaps in the appendix in the revised version. We also emphasized that our work’s core contributions are methodological/conceptual (not tied to specific model versions), and the memorization-capability tradeoff is inherent to LLM pre-training mechanisms, not limited to small-scale models.

II. Applicability Beyond MCQ Formats

We explained the rationale for choosing MCQs as a controlled testbed (clear context-memory target separation for precise quantification; objective correctness evaluation via option labels) and provided evidence for potential extension:
1. **Principle-level applicability**: Removing MCQ options allows reformulation into the entity–attribute–context triplet (Appendix Table 2), proving TrinEval’s core logic is compatible with open-ended tasks.
2. **Preliminary math task experiments**: We tested GSM8K, with 1282/1319 (Qwen3-Max) and 1192/1319 (DeepSeek-R1) questions qualifying for TrinEval reformulation. We will extend this work once robust open-ended evaluation metrics are available.
3. **Future extension plans**: We outlined adaptations for math reasoning (e.g., triplet of “problem entity–calculation attribute–context constraint”) and code generation (e.g., triplet of “function entity–functional attribute–input context”) tasks, as raised by reviewer rvEY (Q2).

## 2. Concerns and Further Discussion of Memorization
**Reviewers KGaV (W4), Nv1h (W4), and rvEY (Q1/Q3) questioned: (1) the ambiguity of “memorization” definition. (2) The discussion of mechanism behind the core finding: “the more LLMs memorize, the worse they perform”.**

I. Clear Definition of Memorization

We reaffirmed that memorization in our work is defined as **verbatim reproduction of content** (line 67 of the paper). For MCQs, we operationalize it as extracting options from the question stem (using Min-K% on option tokens, which has statistical significance), while for open-ended tasks, it can be extended to question-answer matching (we clarified why single-token answer labels in MCQs make this unfeasible). We also noted that modifying question phrasing disrupts memorization triggers, aligning with prior work (Sainz et al., 2023; Carlini et al., 2022).

II. Mechanism of the “Memorization-Performance Trade-Off”

We provided a clear hypothesis for why “more memorization leads to worse performance” (rvEY Q1):
1. **Embedding space evidence**: Memorized questions are sparsely distributed in the embedding space (overfitting to surface token sequences), while capability-related questions form tight semantic clusters (robust conceptual encoding).
2. **Generalization failure**: Rote memorization relies on shallow, context-specific patterns; slight perturbations (via TrinEval reformulation) cause the model to fail, whereas genuine capability enables flexible knowledge application.
3. **Alignment with prior work**: Diverse training rephrasing improves generalization, while fixed-format training reinforces superficial recall, confirming LLMs’ tendency toward rote learning.
Still, we would argue that knowledge learning in LLMs is a continuous spectrum rather than a strict binary.

III. Training Strategies for LTM-Like Knowledge

In response to rvEY (Q3), our findings suggest that **semantic variability in training** can encourage LTM-like representations:
- Exposing models to diverse rephrasings of the same concept reduces overfitting to surface patterns.
- Avoiding repeated verbatim exposure prevents STM-like memorization.

This direction aligns with our future work on improving LLM knowledge robustness.

---

### Meta-Review · Area_Chair_4b8v · 2026-01-05

**Summary:**

This work examines whether language models answer multiple-choice questions through memorization by reformulating such questions. The results reveal a counterintuitive and interesting finding: LLMs perform worse on memorized questions than on non-memorized ones.

Reviewers agree that the research addresses an important and interesting issue: investigating the impact of benchmark contamination on LLM evaluation and attempting to distinguish between rote memorization and genuine understanding.

Reviewers have raised the following main concerns:

1. Insufficient Contribution: The study is narrowly focused on multiple-choice questions, and the models selected are relatively outdated, limiting the generalizability of the conclusions to current models.

2. Dependence on Another LLM: The TrinEval framework relies on another LLM (such as GPT-4) to rephrase questions. This introduces the risk of model-specific bias, as the rephrasing process itself may be influenced by the characteristics of the model used.

**Reviewer Concerns:**

In response, the authors have included additional models, addressing concerns about the generalizability of the conclusions. However, the narrow focus on multiple-choice questions remains an unresolved issue for the reviewers.

**Reviewer Scores:**

Given that the reviewers maintain a consistent weak rejection stance and unresolved concerns persist, they are inclined to retain their current negative evaluations.

---

### Decision · Program_Chairs · 2026-01-26

Reject